# Transcriptional regulation of cyclophilin D by BMP/Smad signaling and its role in osteogenic differentiation

**Rubens Sautchuk[1], Brianna H Kalicharan[1], Katherine Escalera-Rivera[1], Jennifer H Jonason[1,2], George A Porter[3], Hani A Awad[1,4], Roman A Eliseev[1,2,5]***

[1]Center for Musculoskeletal Research, University of Rochester, Rochester, United States; [2]Department of Pathology, University of Rochester, Rochester, United States; [3]Department of Pediatrics, Division of Cardiology, University of Rochester, Rochester, United States; [4]Department of Biomedical Engineering, University of Rochester, Rochester, United States; [5]Department of Pharmacology & Physiology, University of Rochester, Rochester, United States

**\*For correspondence:**
roman_eliseev@urmc.rochester.edu

**Competing interest:** The authors declare that no competing interests exist.

**Abstract** Cyclophilin D (CypD) promotes opening of the mitochondrial permeability transition pore (MPTP) which plays a key role in both cell physiology and pathology. It is, therefore, beneficial for cells to tightly regulate CypD and MPTP but little is known about such regulation. We have reported before that CypD is downregulated and MPTP deactivated during differentiation in various tissues. Herein, we identify BMP/Smad signaling, a major driver of differentiation, as a transcriptional regulator of the CypD gene, *Ppif*. Using osteogenic induction of mesenchymal lineage cells as a BMP/Smad activation-dependent differentiation model, we show that CypD is in fact transcriptionally repressed during this process. The importance of such CypD downregulation is evidenced by the negative effect of CypD 'rescue' via gain-of-function on osteogenesis both in vitro and in a mouse model. In sum, we characterized BMP/Smad signaling as a regulator of CypD expression and elucidated the role of CypD downregulation during cell differentiation.

## Editor's evaluation

This study provides evidence of cyclophilin-D transcriptional regulation of osteoblast differentiation and offers new insights into the underlying mechanisms identifying BMP/Smad signaling as a major mediator. The study has significant relevance and implication to bone health and offers a strong foundation for future work to decipher its unique role in different cells of the mesenchymal lineage and relevance to various bone-related diseases.

## Introduction

The mitochondrial permeability transition pore (MPTP) is a non-selective high-conductance channel within the inner mitochondrial membrane (IMM). MPTP opening leads to the increased permeability of the IMM and entry of solutes up to 1.5 kDa of molecular mass, that is, mitochondrial permeability transition (*Bernardi et al., 2015*). Although the MPTP molecular identity remains debatable, the ATP synthase converges as a potential locus. The consequences of MPTP opening span from physiological events such as regulation of synthasome assembly, oxidative phosphorylation (OxPhos), membrane potential ($\Delta\phi$m), ROS-induced ROS release, $Ca^{2+}$ homeostasis, and epigenetic regulation (*Feissner et al., 2009*; *Bernardi and von Stockum, 2012*; *Beutner et al., 2017*; *Boyman et al., 2019*) to patho-physiological processes associated with sustained pore opening, including mitochondrial dysfunction,

mtDNA release, inflammation, and cell death. Such pathophysiological processes are observed in cancer, aging, injury, and degenerative diseases (*Du and Yan, 2010*; *Giang et al., 2013*; *Alavian et al., 2014*; *Bernardi et al., 2015*; *Warne et al., 2016*; *Rottenberg and Hoek, 2017*). Among some important physiological events under MPTP control is OxPhos since MPTP opening leads to IMM depolarization, proton motive force dissipation, and ATP production decrease. For differentiating cells switching from a glycolytic pathway to higher OxPhos activity, MPTP opening can be particularly detrimental by impairing lineage commitment, differentiation, and functioning of differentiated cells (*Eliseev et al., 2007*; *Hom et al., 2011*; *Lingan et al., 2017*). Currently, several MPTP opening effectors have been described, but the mitochondrial matrix protein cyclophilin D (CypD) remains the only genetically proven opener of the MPTP (*Briston et al., 2019*).

Encoded by the nuclear gene *PPIF*, CypD is a chaperone protein – peptidyl-prolyl cis-trans isomerase F – thought to be involved in protein folding. However, very few reports demonstrate such an activity of CypD (*Porter and Beutner, 2018*). Regarded as the master regulator of MPTP, CypD is imported into mitochondria where it can actively bind to the mitochondrial ATP synthase, decreasing the threshold for pore opening and increasing opening probability. The exact molecular mechanism by which CypD regulates the MPTP and mitochondrial function remains unclear. A range of post-translation modifications, such as acetylation and phosphorylation, is known to exert regulatory properties on CypD activity, and consequently on MPTP opening (*Gutiérrez-Aguilar and Baines, 2015*; *Porter and Beutner, 2018*; *Amanakis and Murphy, 2020*). Additionally, it has been shown that CypD also functions as a scaffold protein, being able to cluster various structural and signaling molecules together to modify mitochondrial physiology and bioenergetic response (*Eliseev et al., 2009*; *Porter and Beutner, 2018*). Although several aspects affecting CypD physiological regulation and its interplay with mitochondrial activity have been elucidated, how the CypD gene is transcriptionally regulated has yet to be described.

CypD expression and activity are cell-specific (*Laker et al., 2016*) and can be regulated during differentiation. As cells become more active and increase their oxygen consumption, higher ROS levels are produced partially due to an increase in the electron flow within the respiratory chain (*Turrens, 1997*; *Barja, 1999*; *Dröse and Brandt, 2012*; *Suski et al., 2018*). Higher ROS levels due to OxPhos activation can lead to deleterious effects including opening of the MPTP and consequently blunting of mitochondrial function (*Carraro and Bernardi, 2016*). To inhibit MPTP opening, it is beneficial for differentiating cells moving toward a higher OxPhos usage, to downregulate CypD expression and/or activity to be better protected against oxidative stress and support OxPhos. Such an inhibition due to downregulation of CypD has been reported during neuronal and cardiomyocyte differentiation (*Eliseev et al., 2007*; *Hom et al., 2011*; *Lingan et al., 2017*). Conversely, induced pluripotent stem cell reprograming is accompanied by a metabolic shift from OxPhos to glycolysis and not surprisingly, by transient MPTP opening and CypD expression upregulation (*Ying et al., 2018*).

The bone morphogenic protein (BMP) signaling pathway is an important driver of cell differentiation and maturation. BMPs are shown to regulate early processes from embryogenesis and development to adult tissue homeostasis. BMPs are not bone exclusive, the ubiquitous expression of BMP underscores its importance for organogenesis and maintenance in several tissues, such as neurological, ophthalmic, cardiovascular, pulmonary, gastrointestinal, urinary, and musculoskeletal systems (*Wang et al., 2014*; *Nickel and Mueller, 2019*). Upon BMP receptor activation and trans-phosphorylation, receptor regulated Smads (R-Smads) such as Smad1/5/8 are recruited and phosphorylated. Phospho-Smad1/5/8 binds to Smad4 forming a heterodimeric complex which is subsequently translocated into the nucleus to activate or repress the expression of its target genes. Osteogenesis and osteoblast (OB) differentiation are controlled at least in part via BMP/Smad canonical signaling. During osteogenic differentiation, activation of the transcription factor Runx2 is mediated by the BMP/Smad signaling pathway initiating a cascade of downstream osteogenic marker genes expression (*Bianco and Robey, 2015*). Our lab and others have demonstrated that bone marrow stromal (a.k.a. mesenchymal stem) cells (BMSCs) shift their bioenergetic profile toward OxPhos during OB differentiation (*Shum et al., 2016a*; *Feigenson et al., 2017*; *Colaianni et al., 2018*; *Shares et al., 2018*; *Yu et al., 2019*; *Smith and Eliseev, 2021*). Since OB function involves bone matrix deposition, an energy-consuming process highly dependent on mitochondrial bioenergetics and OxPhos activity, we hypothesized that MPTP inhibition via CypD transcriptional downregulation is mediated by BMP/Smad signaling and required to sustain OxPhos function and support osteogenic differentiation.

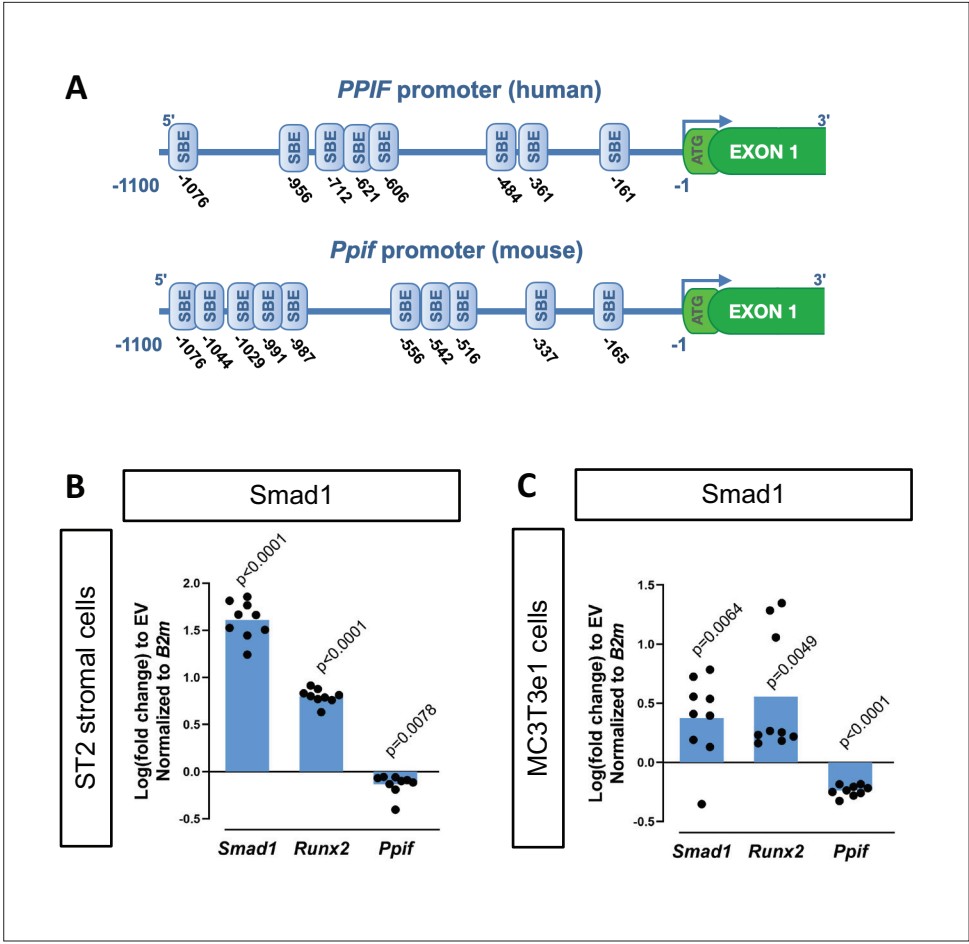

**Figure 1.** Cyclophilin D (CypD) gene, *Ppif*, promoter contains multiple Smad-binding elements (SBEs) and Smad1 overexpression leads to downregulation of CypD. (**A**) Several SBEs were found on both human (*PPIF*) and mouse (*Ppif*) CypD gene promoter. ST2 or MC3T3e1 cells were transfected with pCMV-Smad1 vector or empty vector (EV) control. (**B**) and (**C**) Real-time RT-PCR data demonstrating the efficiency of Smad1 transfection and that Smad1 overexpression upregulated *Runx2* and downregulated *Ppif* mRNA expression. Plot shows the actual data points (biological replicates), calculated means and p value vs. EV controls determined by an unpaired t-test.

## Results

### BMP/Smad signaling is a transcriptional repressor of *Ppif* gene

To determine potential mechanisms involved in CypD regulation on a transcriptional level, we performed an in silico analysis of the CypD gene, *Ppif*, promoter for transcription factor (TF)-binding sites using PROMO online platform (*Messeguer et al., 2002*). We found multiple BMP-dependent Smad-binding elements (SBEs) in the 1.1 kb 5' upstream region of both human *PPIF* and mouse *Ppif* genes (*Figure 1A*). BMP/Smad signaling pathway is an important driver of differentiation in various lineages including osteogenic lineage. Therefore, to test if BMP/Smad signaling regulates CypD gene transcription, we transfected mouse osteogenic bone marrow-derived and mesodermal origin ST2 cells, or calvaria-derived and neural crest-origin MC3T3-e1 cells with pCMV-Smad1 expression vector. The efficiency of Smad1 transfection was confirmed by real-time RT-qPCR. We observed that CypD mRNA was downregulated after Smad1 overexpression, whereas osteogenic marker *Runx2*, a readout of BMP/Smad activity, was upregulated (*Figure 1B–C*).

To confirm interaction of BMP-dependent Smad1 with the *Ppif* promoter, we performed chromatin immunoprecipitation (ChIP) DNA-binding assay using nuclear fractions from ST2 cells supplemented with BMP2 or vehicle control for 24 hr. PCR analysis of the reversed cross-linked protein-DNA immuno-precipitate complexes was done (*Figure 2A*) using primers to amplify the distal region within the *Ppif* promoter containing multiple SBEs (*Figure 2B*). Smad1-*Ppif* interaction was found to be present at

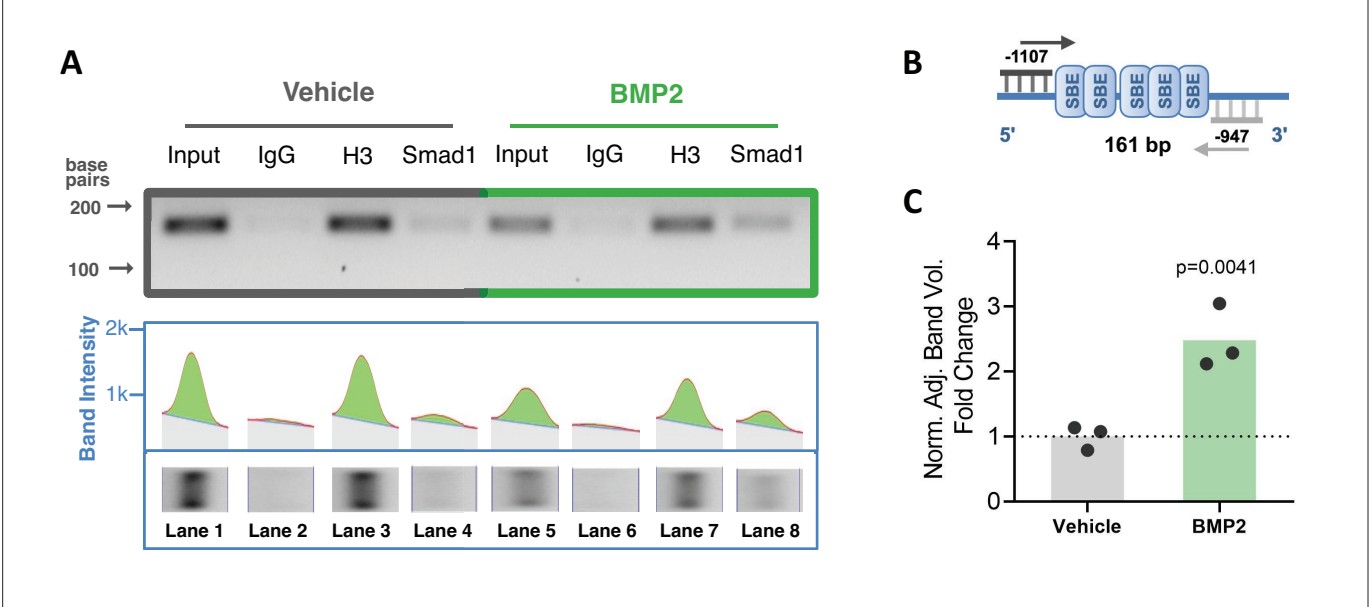

**Figure 2.** Smad1 binds *Ppif* promoter and bone morphogenic protein (BMP) signaling promotes Smad1 interaction with *Ppif* promoter. Chromatin immunoprecipitation (ChIP) assay of nuclear fractions from ST2 cells incubated either in the presence of 50 ng/ml BMP2 for 24 hr, or vehicle. (**A**) PCR analysis of the ChIP assay was performed using primers to amplify the distal Smad-binding elements (SBE)-containing region within the *Ppif* promoter (**B**) .Positive control histone H3 signal showing that proper DNA fragmentation was achieved (**A**). (**C**) Band density quantification for Smad1 immunoprecipitation in both conditions was adjusted by background subtraction and normalized to total DNA input. Unspecific signal from IgG band was also subtracted from Smad1-specific signal. Plot shows the actual data points representing the mean of two technical replicates from three independent experiments, calculated means and p value determined by an unpaired t-test.

The online version of this article includes the following source data for figure 2:

**Source data 1.** Chromatin immunoprecipitation (ChIP) assay of nuclear fractions from ST2 cells incubated either in the presence of 50 ng/ml bone morphogenic protein 2 (BMP2) for 24 hr, or vehicle.

**Source data 2.** PCR analysis of the ChIP assay for Smad binding of *Ppif* promoter.

low levels in the vehicle-treated control cells. This finding is consistent with the fact that intrinsic levels of BMP activity are present even in undifferentiated osteogenic cells. Importantly, BMP2 treatment significantly induced Smad1-*Ppif* interaction. Band density quantification revealed 2.5-fold increase of Smad1-*Ppif* interaction in the BMP2-treated samples (***Figure 2C***). These results demonstrate that Smad1 binds the CypD gene (*Ppif*) promoter in BMP-dependent manner.

To characterize the functionality of the SBEs found within the *Ppif* promoter, we subcloned the 1.1 kb mouse *Ppif* promoter into the pGL4.10 luciferase reporter construct (***Figure 3A***). We then co-transfected ST2 cells with the above promoter-reporter construct and either pCMV-Smad1 or pCMV empty vector (EV) and analyzed the luciferase signal. ***Figure 3C*** shows that Smad1 significantly decreased the luciferase signal from the 1.1 kb *Ppif*-luc promoter-reporter. Furthermore, a promoter bashing approach was used and five other various length promoter constructs were generated (***Figure 3— figure supplement 1A***): –0.37 to –0.1 kb, or –0.62 to –0.45 kb, or –1.1 to –0.95 kb, or –0.62 to –0.1 kb, or –1.1 to –0.45 kb deletion mutants that correspond to a cluster of the two most proximal (P), three middle (M), five most distal (D), five middle + proximal (M+P), and eight distal + middle (D+M) SBEs, respectively. We co-transfected ST2 cells with the above promoter-reporter constructs and either pCMV-Smad1 or pCMV-EV and analyzed the luciferase signal for all our constructs. Middle and distal regions did not show any luciferase signal difference when compared to EV-transfected cells, whereas proximal, M+P, and D+M rescued *Ppif* activity (***Figure 3—figure supplement 1B***). Thus, only the full-length 1.1 kb *Ppif*-luc reporter showed *Ppif* activity repression upon Smad1 transfection (***Figure 3C***). Knowing that the full-length *Ppif* promoter is the actual region controlling the gene activity and that Smads work by forming oligomeric structures binding various regions of the promoter (***Massagué et al., 2005***; ***Wang et al., 2014***), we considered the data using the 1.1 kb *Ppif*-luc reporter as the

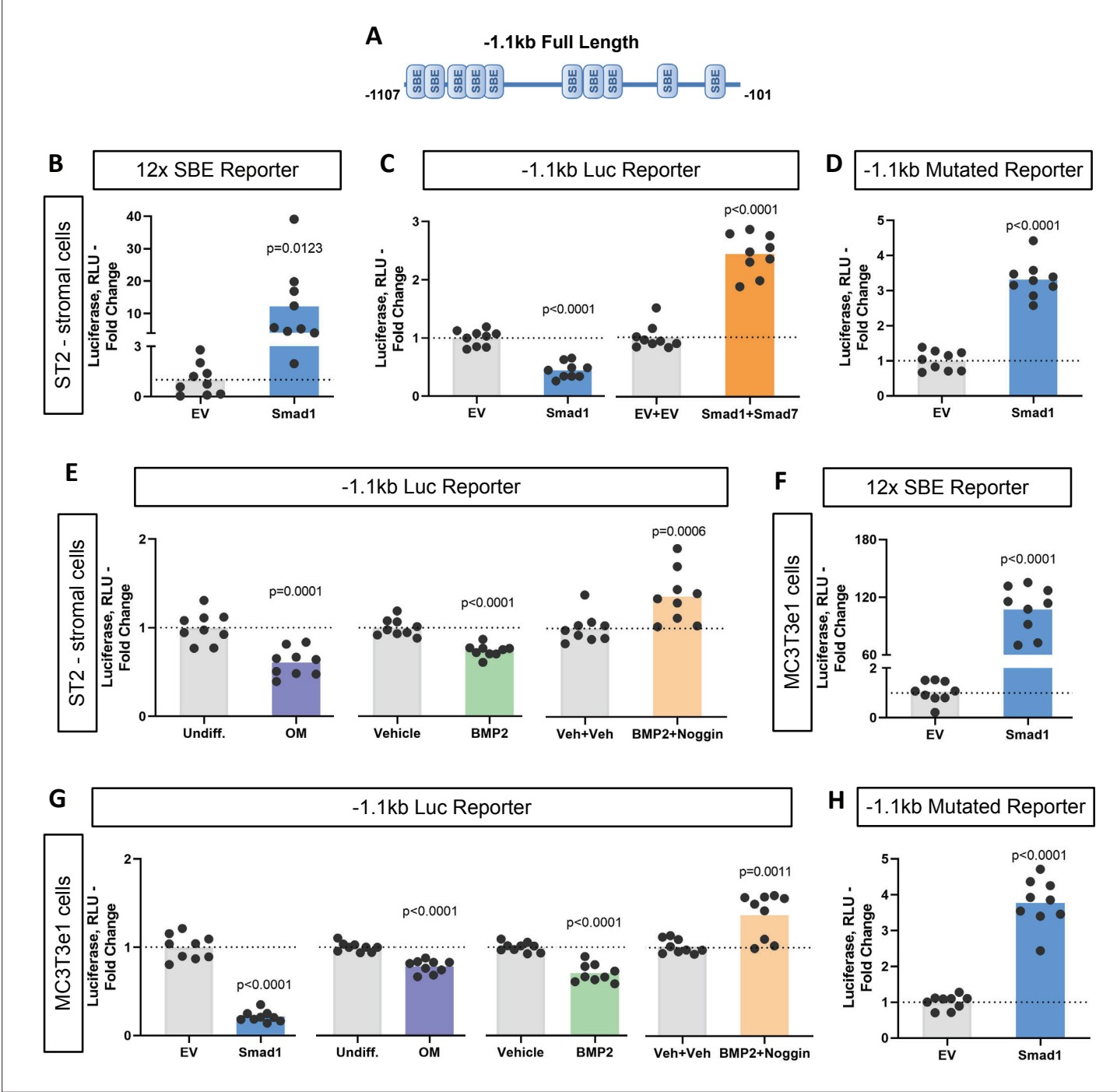

**Figure 3.** Bone morphogenic protein (BMP)-dependent Smad1 transcriptionally represses *Ppif* promoter activity. Dual luciferase reporter assay was performed on either ST2 or MC3T3e1 cells 48 hr after luciferase (Luc) reporter transfection. (**A**) Diagram: 1.1 kb full-length *Ppif* promoter region containing several Smad-binding elements (SBEs) was cloned into the pGL4.10 vector. (**B**) and (**F**) pCMV-Smad1 co-transfection highly activated the BMP/Smad signaling reporter 12xSBE. (**C**) pCMV-Smad1 co-transfected with the 1.1 kb *Ppif* full-length luc reporter downregulated the luciferase signal. Inhibitory Smad7 rescued *Ppif* promoter activity. (**D**) and (**H**) pCMV-Smad1 co-transfected with the 1.1 kb SBE-mutated *Ppif* full-length luc reporter rescued luciferase signal. (**E**) Osteogenic media or 50 ng/ml BMP2 was used to activate BMP/Smad signaling. The BMP inhibitor, Noggin, rescued *Ppif* promoter activity. (**G**) MC3T3e1 cells showed similar effects on *Ppif* promoter activity after BMP/Smad signaling activation. Plot shows the actual data points (biological replicates), calculated means and p value vs. EV controls determined by an unpaired t-test.

The online version of this article includes the following figure supplement(s) for figure 3:

**Figure supplement 1.** Bone morphogenic protein (BMP)-dependent Smad1 transcriptionally represses only full-length *Ppif* promoter activity.

most relevant. Of note, in all these experiments, activation of Smad1 signaling following transfection was confirmed by the increase in the activity of 12xSBE, a BMP/Smad signaling luciferase reporter (*Figure 3B*). In sum, our data indicate that BMP/Smad signaling is a transcriptional repressor of CypD.

To further delineate the role of BMP/Smad signaling in *Ppif* repression, we used the inhibitory Smad7 which rescued *Ppif* promoter activity in Smad1-transfected cells (*Figure 3C*). Smad7 competitively inhibits the interaction of R-Smads to the cytoplasmic domain of their respective cell receptors, therefore preventing R-Smad recruitment and phosphorylation, and consequently nuclear translocation. Binding specificity was confirmed using a 1.1 kb *Ppif*-luc reporter designed to introduce point mutations in the SBEs and therefore, prevent R-Smads binding. The 1.1 kb mutated *Ppif*-luc reporter co-transfected with pCMV-Smad1 rescued *Ppif* promoter activity (*Figure 3D*). This result was similar to the conditions meant to inhibit BMP/Smad signaling (*Figure 3—figure supplement 1C*), confirming the specificity of SBEs in the *Ppif* promoter. Additionally, cells transfected with the 1.1 kb *Ppif*-luc reporter were either treated with BMP2 with or without the BMP inhibitor Noggin or induced in osteogenic media and assayed for luciferase signal. A dose-response assay revealed that BMP2 at 50 ng induced the highest suppression level of *Ppif* activity (*Figure 3—figure supplement 1D*). BMP2 and osteogenic media downregulated the luciferase signal, whereas Noggin rescued *Ppif* activity (*Figure 3E*). These results were also confirmed in MC3T3e1 cells (*Figure 3F–H*). However, the mouse myogenic C2C12 cell line showed opposite effects under BMP-dependent Smad activation (*Figure 3—figure supplement 1F*). These findings show that *PPIF* activity is also regulated by Smad-dependent signaling in myogenic cells but suggest that the direction of *PPIF* regulation is tissue specific. Taken together, these data support our hypothesis that BMP/Smad signaling transcriptionally represses *Ppif* promoter activity in osteogenic cells regardless of their embryonic origin and consequently downregulates CypD expression during osteogenic differentiation.

## Decreased CypD expression and MPTP activity during osteogenic differentiation

BMP/Smad signaling is a major driver of differentiation. It is particularly important for osteogenic lineage. During osteogenic differentiation, cell energy metabolism shifts toward OxPhos as was shown by us and others (*Shum et al., 2016a*; *Feigenson et al., 2017*; *Colaianni et al., 2018*; *Shares et al., 2018*; *Yu et al., 2019*; *Smith and Eliseev, 2021*). Since a higher OxPhos activity, as seen in differentiated OBs, is described to produce more ROS (*Barja, 1999*; *Dröse and Brandt, 2012*; *Suski et al., 2018*; *Turrens, 1997*; *Figure 4—figure supplement 1A*), potentially leading to oxidative stress and higher probability of the MPTP opening, it is beneficial for actively respiring cells such as OBs to decrease MPTP activity, for example, by downregulating CypD. We therefore measured CypD expression and MPTP activity in several osteogenic cell types. *Figure 4* shows that primary BMSCs isolated from mouse long bones, mouse bone marrow-derived osteogenic cell line ST2, and mouse calvaria-derived osteogenic cell line MC3T3-e1 underwent OB differentiation confirmed by alkaline phosphate and alizarin red staining (*Figure 4A–B*) and upregulation of osteogenic markers *Alp* and *Bglap*.(*Figure 4C–E*). In accordance with our hypothesis, osteogenic differentiation was accompanied by downregulation of CypD gene, *Ppif*, mRNA expression in all these cell types (*Figure 4C–E*). To confirm that this effect is not species-specific, we analyzed human BMSC RNAseq dataset from our previous publication (*Shum et al., 2016a*) and observed downregulation of CypD gene, *PPIF*, mRNA expression in osteoinduced cells when compared to undifferentiated cells (*Figure 4—figure supplement 1B*). Protein content of CypD correspondingly decreased during OB differentiation as measured with western blot (*Figure 4—figure supplement 2*). We then measured MPTP activity using calcein-cobalt assay and flow cytometry (*Figure 4F*). In this assay, cells are incubated with calcein-AM in the presence of $CoCl_2$ which quenches cytosolic but not mitochondrial calcein unless mitochondria are permeabilized due to MPTP opening. The increase in the calcein signal indicates higher resistance to MPTP opening and decreased pore activity (*Petronilli et al., 1999*). The assay showed that calcein signal increased at day 14 of OB differentiation when compared to day 0 indicating lower MPTP activity (*Figure 4F*). No changes were detected in the total mitochondrial mass during OB differentiation as labeled by nonyl acridine orange (*Figure 4G*), therefore confirming that calcein signal increase is in fact caused by a lower pore activity and not by an increased mitochondrial compartment. This is consistent with our previous report showing that mitochondrial mass and mtDNA do not increase during OB differentiation (*Shum et al., 2016a*; *Shares et al., 2018*). Altogether, these data

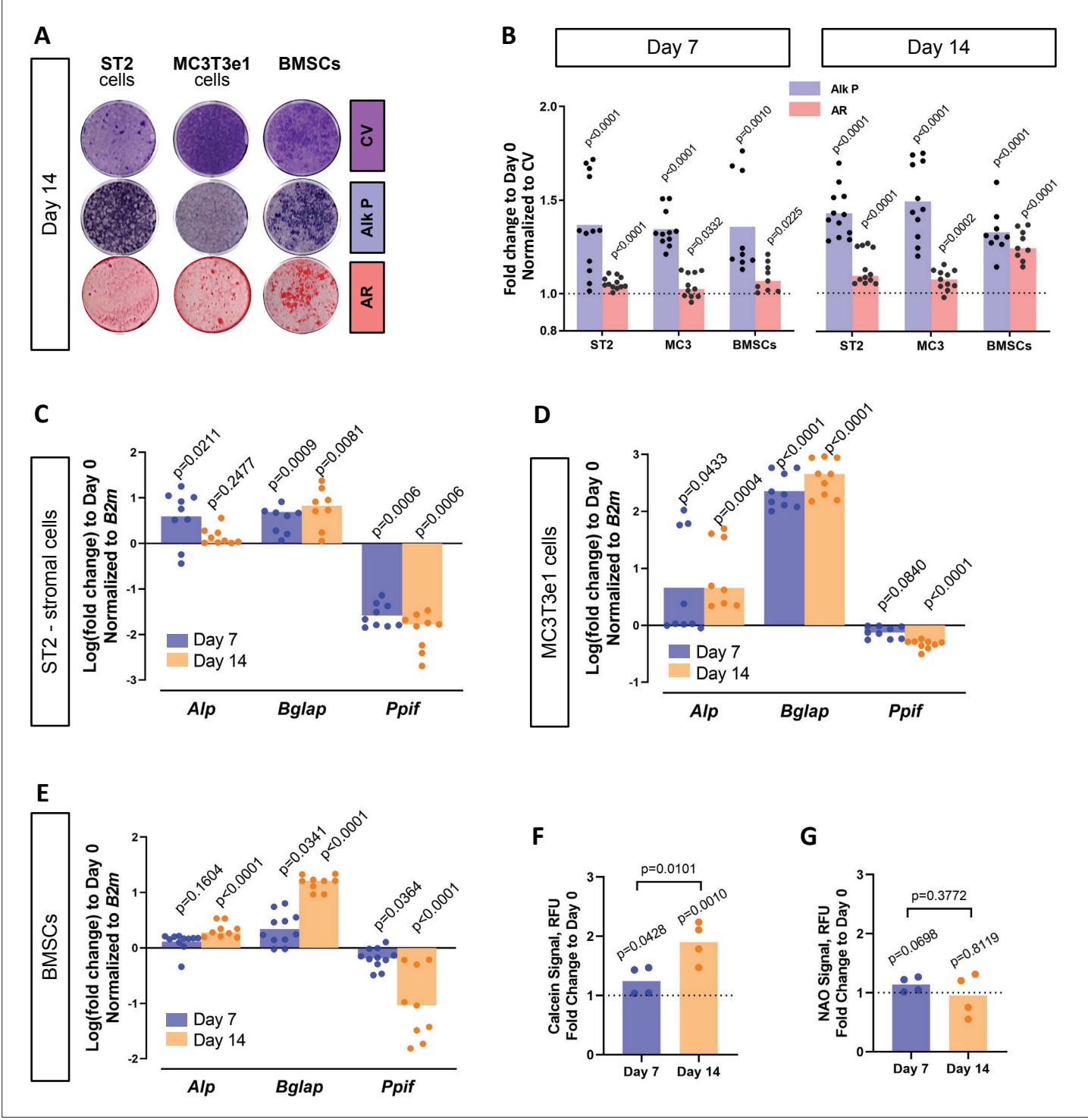

**Figure 4.** Cyclophilin D (CypD) expression and mitochondrial permeability transition pore (MPTP) activity are downregulated during osteogenic differentiation. ST2 stromal cells, MC3T3e1, or mouse bone marrow stromal cells (BMSCs) were cultured in osteogenic media; RNA collected, and staining done at day 0, day 7, and day 14. (**A**) Staining representative image at day 14 for crystal violet (CV) a proxy of total cell content, alkaline phosphatase (Alk P), and alizarin red (AR). (**B**) Staining quantification confirming osteogenic commitment. (**C**), (**D**), and (**E**) Real-time RT-PCR data: osteogenic markers (*Alp* and *Bglap*) are upregulated, whereas *Ppif* mRNA expression is downregulated. (**F**) BMSCs subjected to calcein-cobalt assay showed decreased MPTP activity upon osteogenic differentiation. (**G**) BMSCs stained with nonyl acridine orange (NAO) show no difference in mitochondrial mass. Plot shows the actual data points (biological replicates), calculated means, and p value vs. D0 determined by an unpaired t-test.

The online version of this article includes the following source data and figure supplement(s) for figure 4:

*Figure 4 continued on next page*

*Figure 4 continued*

**Figure supplement 1.** Human bone marrow stromal cell (BMSC) osteogenic differentiation increases mitochondrial ROS while downregulating *PPIF* relative expression.

**Figure supplement 2.** Mouse bone marrow stromal cells (BMSCs) osteogenic differentiation downregulates cyclophilin D (CypD) protein expression.

**Figure supplement 2—source data 1.** Mouse bone marrow stromal cells (BMSCs) were cultured in osteogenic media and protein lysates collected at day 0 and day 14.

indicate that CypD expression is downregulated and MPTP activity is decreased during osteogenic differentiation.

## Smad-mediated regulatory effect on CypD is independent from osteogenic signaling downstream of Smad1

Osteogenic differentiation is a complex process that involves coordination and crosstalk of several signaling pathways. BMP/Smad signaling is a potent driver of osteogenic differentiation in BMSCs. Once the BMP response is activated, other signaling molecules can influence Smad activity and osteogenic differentiation. For instance, after Smad1-Smad4 complex nuclear translocation, its dephosphorylation is accompanied by the dissociation of the complex and export to the cytoplasm. Smad4 is then believed to interact with β-catenin and TCF/LEF1 forming a TF activation complex, now under the control of Wnt signaling. Additionally, we found TCF/LEF1 regulatory elements in the 1.1 kb *Ppif* promoter region (*Figure 5—figure supplement 1A*). To determine the role of Wnt signaling in *Ppif* regulation, we transfected the 1.1 kb *Ppif*-luc reporter in ST2 cells and performed a dose-response assay using recombinant Wnt3a. Even Wnt3a concentrations of 25 ng and above, previously reported to activate OB differentiation and mitochondrial function in osteogenic cells (*Smith and Eliseev, 2021*), showed no effect on *Ppif* activity (*Figure 5—figure supplement 1B*). We also investigated the role of BMP/Smad signaling in CypD gene, *Ppif*, repression in a cell line where osteogenic signaling downstream of BMP/Smad is arrested. We previously reported that the human osteosarcoma (OS) cell line 143b cannot undergo osteogenic differentiation due to Runx2 proteasomal degradation (*Shapovalov et al., 2010*). Even though *Runx2* mRNA expression is upregulated in osteoinduced 143b-OS cells, the osteogenic pathway cascade is blocked by this post-translational regulation. We confirmed the 143b-OS cells' inability to fully differentiate upon BMP2 osteogenic induction by comparing their mRNA expression of osteoblastic differentiation markers to that of human non-cancerous osteoblastic cell line, hFOB (*Figure 5A*). While hFOB cells upregulate *ALP*, *IBSP*, and *BGLAP*, 143b cells fail to upregulate these osteogenic markers. As seen in the ST2 and MC3T3e1 cell lines, pCMV-Smad1 transfection led to downregulation of CypD mRNA expression in 143b-OS cells (*Figure 5B*) and decreased *Ppif* promoter activity when co-transfected with the full-length 1.1 kb *Ppif*-luc reporter (*Figure 5C*). Altogether, our data strongly indicate that BMP/Smad signaling exerts inhibitory effect on the *Ppif* gene as a direct transcriptional repressor and not as an indirect result of osteogenic differentiation (*Figure 5D*).

## Downregulation of CypD is important for osteogenic differentiation

We previously reported that CypD knock-out (KO) mice, a loss-of-function (LOF) model of the MPTP, present with higher BMSC OxPhos function and osteogenic potential (*Shum et al., 2016b*). Moreover, RNAseq transcriptome analysis showed that CypD KO BMSCs present pro-osteogenic gene signatures (*Shares et al., 2020*). However, it has not yet been established that CypD downregulation is absolutely necessary for OB differentiation. To address this question, we studied the effect of CypD re-expression and CypD/MPTP gain-of-function (GOF) on OB differentiation. It is known that acetylation of CypD at K166 increases CypD activity and MPTP opening and can be mimicked by K166Q mutation (*Alkalaeva et al., 2009*; *Porter and Beutner, 2018*). We therefore used the pCMV6-*caPpif* vector that expresses constitutively active CypD (caCypD) K166Q mutant tagged with c-Myc and DDK-Flag peptides to stably transfect MC3T3e1 cells. Expression of caCypD mutant was confirmed by western blot (*Figure 6A*). Cells expressing caCypD showed impaired osteogenic differentiation as evidenced by lower expression of OB marker genes, *Runx2*, *Alp*, and *Bglap*, at both days 7 and 14 in osteogenic media (*Figure 6B*).

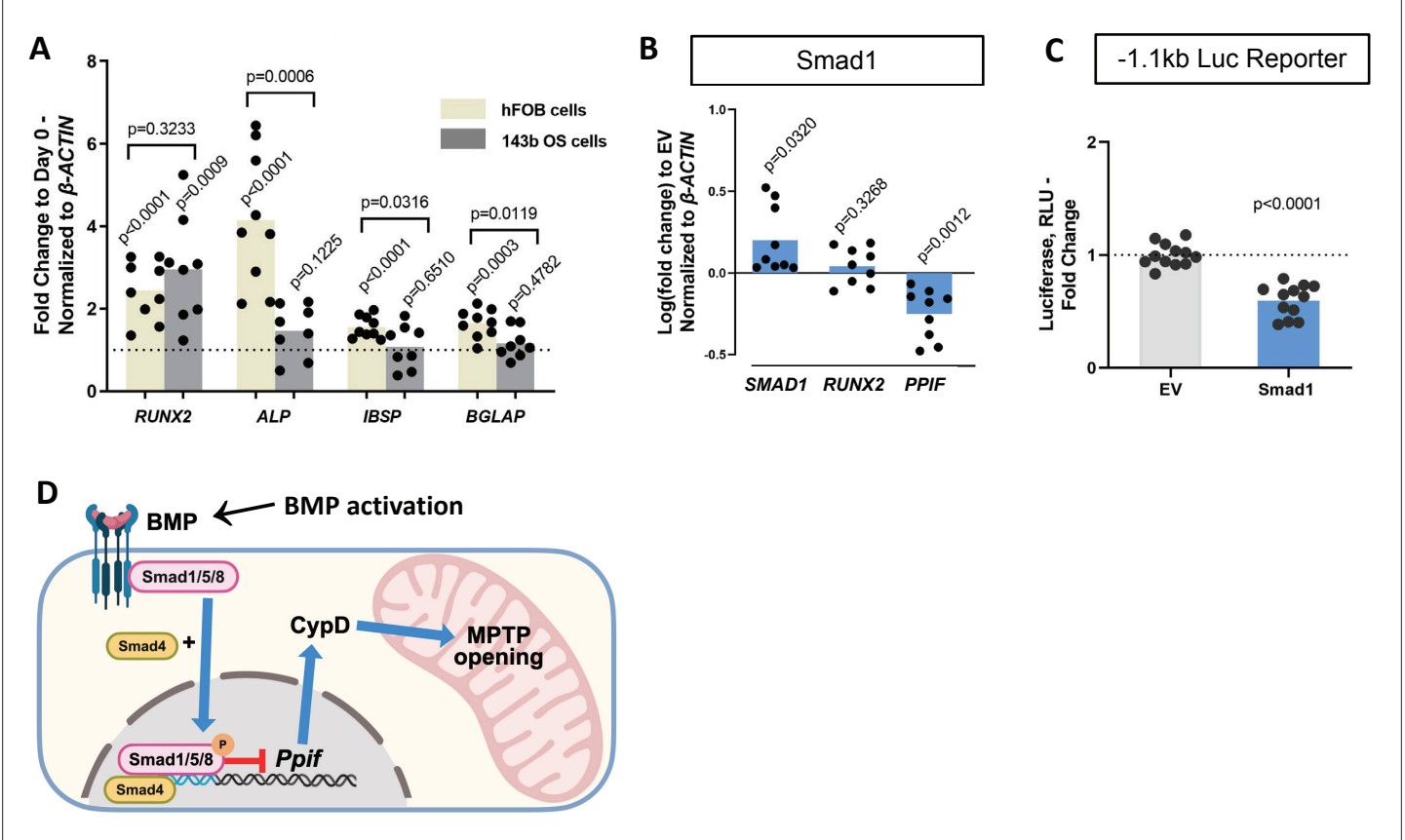

**Figure 5.** Smad1-mediated CypD/*Ppif* repression is not dependent on signaling downstream of BMP/Smad. Immortalized human osteoblastic cells (hFOB) or osteosarcoma cells (143b) were treated with 50 ng BMP2 for 5 days to induce osteogenic differentiation. BMP2 induced early (*RUNX2*) but not late (*IBSP*, *BGLAP*) OB markers in 143b-OS cells, reflecting differentiation-arrested phenotype. (**B**) Real-time RT-PCR data demonstrating the efficiency of Smad1 transfection and that Smad1 overexpression downregulated *Ppif* mRNA expression in 143b cells. (**C**) pCMV-Smad1 co-transfected with the –1.1 kb *Ppif* full-length luc reporter downregulated the luciferase signal. (**D**) Schematic representation of our summary of results. To maintain bone marrow stromal cell (BMSC) commitment to the osteogenic lineage, closure of the mitochondrial permeability transition pore (MPTP) is required, which is achieved by CypD downregulation through Smad1 transcriptional repression of *Ppif* gene. Plots show the actual data points (biological replicates), calculated means, and p value vs. D0 (**A**) or empty vector (EV) controls (**B**) determined by an unpaired t-test. BMP, bone morphogenic protein; CypD, cyclophilin D.

The online version of this article includes the following figure supplement(s) for figure 5:

**Figure supplement 1.** Wnt-activated TCF/Lef1 binding site has no effect on *Ppif* promoter activity.

As mentioned above, CypD LOF mouse models were created before and in our hands demonstrated the beneficial effects of CypD deletion on bone (*Shum et al., 2016b*; *Shares et al., 2020*; *Laura et al., 2020*). However, CypD GOF mouse model was not previously created except for one model that is no longer available (*Baines et al., 2005*). We therefore designed a tissue-specific knock-in mouse model of CypD GOF that expresses the above caCypD mutant in the presence of Cre recombinase. Briefly, *caPpif* cDNA encoding CypD K166Q mutant preceded by the floxed 'Stop' codon and followed by an IRES sequence and eGFP cDNA was inserted into the Rosa26 locus (C57Bl/6 background). This new mouse line was named *R26^caPpif*. The presence of this transgene was confirmed with genotyping (*Figure 6—figure supplement 1*). To achieve OB-specific caCypD expression, we crossed these mice with tamoxifen-inducible 2.3 kb *Col1^CreERt2* mice (final cross is *Col1^CreERt2/+; R26^caPpif/+*). BMSCs were isolated from these skeletally mature 3-month-old male mice, expanded and induced to differentiate in osteogenic media. Addition of 4'-OH-tamoxifen to osteoinduced BMSCs prompted recombination at day 11 (*Figure 6C*), coinciding with *Col1a1* expression time-point during the course of OB differentiation induced in vitro (*Safadi et al., 2009*). Translation of caCypD mutant was confirmed by western blot (*Figure 6D*). Cells were harvested at day 14 of osteoinduction and analyzed. Osteoinduced caCypD-expressing cells presented with decreased osteogenic differentiation and OB markers

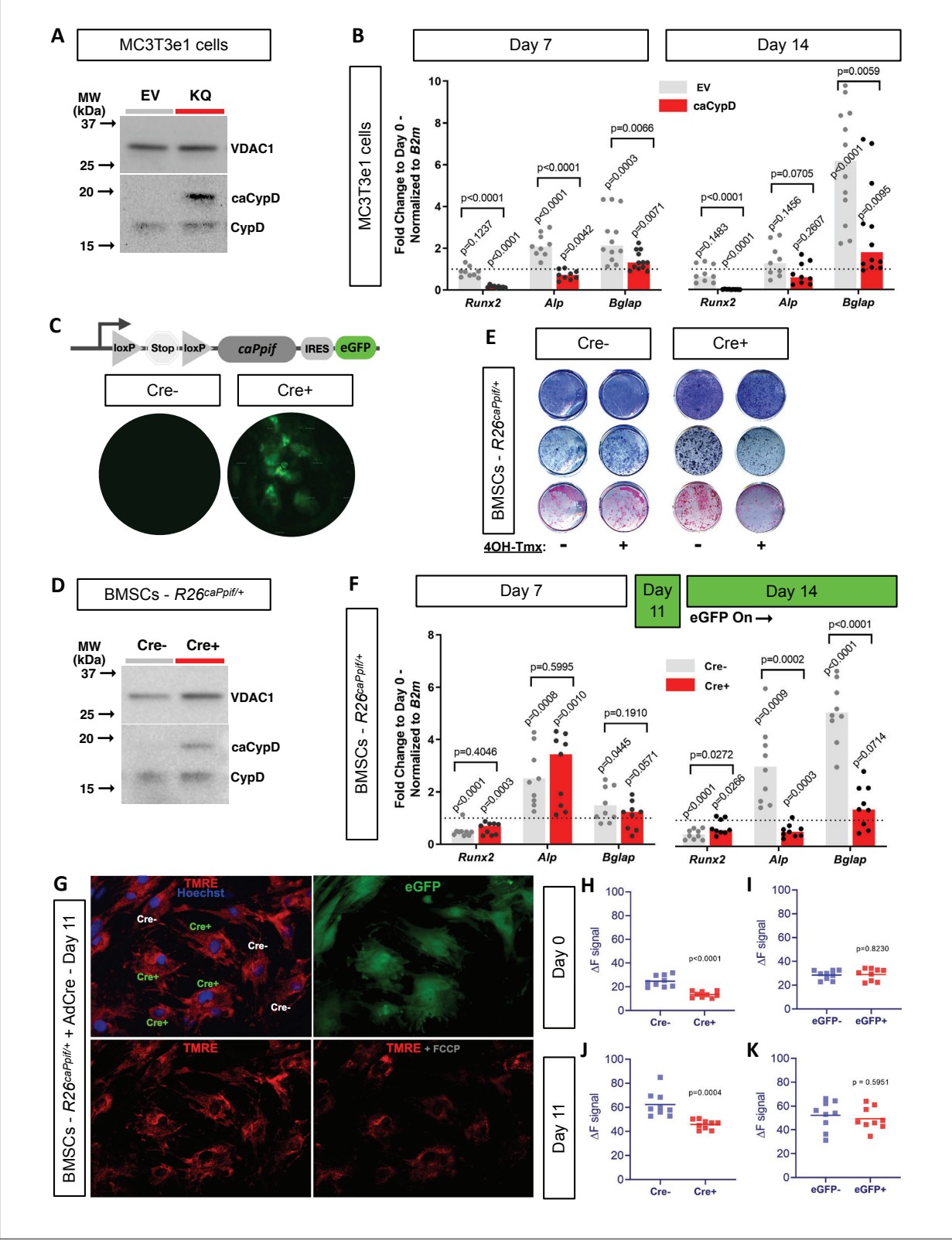

**Figure 6.** Cyclophilin D (CypD) re-expression and gain-of-function impair osteogenic differentiation and mitochondrial function. MC3T3e1 were stably transfected cells with pCMV6-*caPpif* vector to express constitutively active CypD K166Q mutant and thus achieve CypD gain-of-function (GOF). Bone marrow stromal cells (BMSCs) from either osteoblast (OB)-specific, inducible 2.3 kb *Col1^CreERt2^;R26^caPpif/+^* mice or *R26^caPpif/+^* mice were cultured in osteogenic media for 11–14 days. (**A**) Representative western blot of stably transfected MC3T3e1 cells (EV: pCMV6-empty vector; KQ: pCMV6- *caPpif*

*Figure 6 continued on next page*

*Figure 6 continued*

vector; VDAC1: loading control; caCypD: mutant CypD; CypD: endogenous CypD). (**B**) and (**F**) Real-time RT-PCR data of osteogenic markers: CypD GOF MC3T3 cells and OB-specific, inducible 2.3 kb *Col1$^{CreERt2}$;R26$^{caPpif/+}$* BMSCs were incapable to complete OB differentiation. (**C**) Diagram showing the *caPpif* mouse transgene construct. Images show eGFP signal in cell culture confirming recombination induced in vitro. (**D**) Representative western blot of BMSCs from 2.3 kb *Col1$^{CreERt2}$;R26$^{caPpif/+}$* mice. (**E**) Staining representative image at day 14 for crystal violet, alkaline phosphatase, and alizarin red: CypD GOF (tamoxifen-induced Cre$^+$) BMSCs showed decreased OB activity and mineralization capacity. (**G**) Representative images from *R26$^{caPpif/+}$* BMSCs infected with adeno-Cre virus, culture in osteogenic media during 11 days. (**H**) and (**J**) Osteoinduced *R26$^{caPpif/+}$* BMSCs infected with adeno-Cre collected at day 0 or day 11 showed decreased TMRE ΔF signal for infected (Cre$^+$) cells compared to non-infected (Cre$^-$) cells. (**I**) and (**K**) Osteoinduced *R26$^{caPpif/+}$* BMSCs infected with adeno-eGFP collected at day 0 or day 11 showed no difference in TMRE ΔF signal between infected (Cre$^+$) and non-infected (Cre$^-$) cells. Plot shows the actual data points (biological replicates), calculated means, and p value vs. D0 (**B**, **F**) or Cre$^-$ (**H–K**) controls determined by an unpaired t-test.

The online version of this article includes the following source data and figure supplement(s) for figure 6:

**Source data 1.** Original western blot image of stably transfected MC3T3e1 cells probed for cyclophilin D (CypD).

**Source data 2.** Labeled western blot image of stably transfected MC3T3e1 cells probed for cyclophilin D (CypD) (EV: pCMV6-empty vector; KQ: pCMV6-*caPpif* vector; caCypD: mutant CypD; CypD: endogenous CypD).

**Source data 3.** Original western blot image of stably transfected MC3T3e1 cells probed for VDAC1.

**Source data 4.** Labeled western blot image of stably transfected MC3T3e1 cells probed for VDAC1 (EV: pCMV6-empty vector; KQ: pCMV6-*caPpif* vector; VDAC1: loading control).

**Source data 5.** Bone marrow stromal cells (BMSCs) from either osteoblast (OB)-specific, inducible 2.3 kb *Col1$^{CreERt2}$;R26$^{caPpif/+}$* mice or *R26$^{caPpif/+}$* mice were cultured in osteogenic media for 11–14 days.

**Source data 6.** Bone marrow stromal cells (BMSCs) from either osteoblast (OB)-specific, inducible 2.3 kb *Col1$^{CreERt2}$;R26$^{caPpif/+}$* mice or *R26$^{caPpif/+}$* mice were cultured in osteogenic media for 11–14 days.

**Figure supplement 1.** Cyclophilin D (CypD) gain-of-function (GOF) mice genotyping.

**Figure supplement 1—source data 1.** F1 generation for osteoblast (OB)-specific inducible 2.3 kb *Col1$^{CreERt2}$;R26$^{caPpif/+}$* mice genotyping.

**Figure supplement 1—source data 2.** F1 generation for osteoblast (OB)-specific 2.3 kb *Col1$^{CreERt2}$;R26$^{caPpif/+}$* mice genotyping.

**Figure supplement 1—source data 3.** F1 generation for osteoblast (OB)-specific 2.3 kb *Col1$^{CreERt2}$;R26$^{caPpif/+}$* mice were genotyped to confirm the presence of caCypD (K166Q) insert.

**Figure supplement 1—source data 4.** F1 generation for osteoblast (OB)-specific inducible 2.3 kb *Col1$^{CreERt2}$;R26$^{caPpif/+}$* mice were genotyped to confirm the presence of caCypD (K166Q) insert.

when compared to control cells (*Figure 6E and F*). To evaluate the mitochondrial function in the CypD GOF model, we collected BMSCs from *R26$^{caPpif}$* mice and induced recombination through adeno-Cre viral infection. Adeno-eGFP infection was used as negative control for caCypD expression. Transduced BMSCs were induced in osteogenic media for 11 days, stained with the nuclear stain, Hoechst, and the Δψm-dependent stain, TMRE. FCCP, a protonophore, was used as a negative control for TMRE staining (*Figure 6G*). CypD overexpression clearly decreased TMRE signal, and therefore mitochondrial function and OxPhos capacity in infected, eGFP-expressing Cre$^+$, cells compared to non-infected, Cre$^-$, cells in both BMSCs at day 0 and OBs at day 11 (*Figure 6H and J*). Adeno-eGFP showed no important effects on TMRE signal demonstrating that virus infection itself is not the driver of mitochondrial function change (*Figure 6I and K*). Altogether, our results indicate that CypD GOF disrupts mitochondrial function and impairs OB differentiation and thus, downregulation of CypD is an important part of osteogenic program.

## CypD re-expression and GOF reverse the beneficial effects of CypD deletion on bone phenotype in aged mice

To determine the physiological relevance of *PPIF* downregulation driven by BMP/Smad signaling during osteogenesis, we analyzed publicly available RNAseq datasets on bone aging (*Ambrosi et al., 2021*) and on fracture (*Coates et al., 2019*) and found inverse correlation between BMP signaling and *Ppif* expression (*Figure 7—figure supplement 1A*). In particular, downregulation of markers of BMP/Smad signaling was accompanied by significant upregulation of CypD gene, *Ppif*, in aged mouse bones (top left table). On the other hand, upregulation of BMP/Smad signaling during active phase of bone formation (day 14 post-fracture) in mouse fracture callus was accompanied by significant downregulation of *Ppif* (top right table). We also found that in conditions leading to pathological calcification such as osteoarthritis and osteophyte formation (*Dunn et al., 2016*) and cardiac valve

calcification (*Huang et al., 2019*) and associated with upregulated BMP/Smad signaling, *PPIF* expression is decreased (bottom tables).

To further prove the relationship between BMP/Smad signaling and CypD expression, we analyzed bone samples from 3- and 18-month-old mice. These samples were collected before for our previously published work on bone aging (*Shum et al., 2016a*). BMP/Smad signaling is known to be downregulated in aged bone. Using immunofluorescence (IF), we indeed detected decreased levels of BMP2 in bones of aged mice when compared to young mice. Consistent with our hypothesis, levels of CypD correspondingly increased (*Figure 7—figure supplement 1B*, C). In addition, we used bone fracture samples from our previously published work (*Shares et al., 2020*) and performed IF for BMP2 and CypD. Consistent with the published RNAseq dataset described above (*Figure 7—figure supplement 1A*, top right table), BMP2 signal was significantly increased in the callus at day 14 post-fracture when compared to the unfractured bone. This increase likely reflects active bone forming and callus ossification stage. The observed increase in BMP2 signal was accompanied by significant decrease in CypD signal (*Figure 7—figure supplement 1D*, E). Therefore, previously published transcriptomic data and our own studies indicate inverse correlation between BMP signaling and CypD expression.

Higher MPTP activity and CypD expression are shown to be present in aging (*Sun et al., 2016*; *Rottenberg and Hoek, 2017*; *Farr and Almeida, 2018*). Additionally, CypD/MPTP upregulation is involved in several pathophysiological conditions, including premature aging and degenerative diseases (*Bernardi et al., 2015*). Conversely, CypD deletion produces protective effects against several degenerative processes, including bone loss in aged mice (*Shum et al., 2016a*). To confirm our in vitro results and obtain further insight into the CypD/MPTP regulation of bone cell function, we investigated the effects of CypD re-expression and GOF in vivo. For this, we used a viral delivery of the above caCypD K166Q mutant (*caPpif*) in a mouse model where CypD was initially deleted. In our experimental design, OB-specific CypD deletion (2.3 kb *Col1*$^{CreERt2}$;*Ppif*$^{f/f}$) was induced beginning at 2 months of age. These mice were aged to 22 months while maintaining CypD deletion with bimonthly tamoxifen boosts (*Figure 7A*). Cre$^-$ mice (*Ppif*$^{f/f}$) were also injected with tamoxifen and used as controls. To rescue CypD expression and achieve CypD GOF, we designed a CRE-DIO (*Saunders et al., 2012*) viral system carrying *caPpif* and eGFP cDNA. The CRE-DIO system allows expression of the gene of interest only in Cre$^+$ cells in the animal, therefore providing specificity. The AAV2-CRE-DIO-*caPpif*-eGFP vector (AAV-DIO, *Figure 7B*) was introduced via intra-bone marrow injection in the tibiae of the above mice. Contralateral tibiae were injected with sterile phosphate buffered saline (PBS) solution as an intra-mouse control. Two months after the viral infection and recombination, eGFP signal confirming a successful CypD re-expression was present in the AAV-DIO-injected tibiae of Cre$^+$ but not Cre$^-$ mice (*Figure 7C* and *Figure 7—figure supplement 2A*). CypD re-expression and GOF also led to decreased OB function in vivo as confirmed by decreased osteocalcin IF (*Figure 7D*), recapitulating the results we have observed in vitro (*Figure 6E and F*). Since OB function is described to partially regulate osteoclast formation, recruitment and maturation, we analyzed osteoclast activity by tartrate-resistant acid phosphatase (TRAP) staining (*Figure 7E*) and found no differences between the AAV-DIO-injected tibiae of Cre$^+$ and Cre$^-$ mice. An important observation in this experimental design is that OB-specific CypD deletion is in fact protecting against bone loss in aging, corroborating our previous results when we investigated the effects of CypD global deletion in mice (*Shum et al., 2016b*). For instance, biomechanical testing showed increased torsional rigidity in control tibiae of Cre$^+$ (*Ppif*$^{-/-}$) mice when compared to Cre$^-$ (*Ppif*$^{+/+}$) mice (*Figure 7H*), indicative of an increased bone strength. Bone volume fraction measured with micro-computed tomography (μCT), a strong predictor of bone strength (*Nazarian et al., 2008*), provided further evidence that OB-specific CypD deletion effectively protects against bone degeneration in aging (*Figure 7G*). Although bones with AAV-DIO-delivered CypD re-expression showed decreased cortical thickness and torsional rigidity when compared to their respective contralateral PBS-injected tibiae (*Figure 7F and H*), we further analyzed our data using data normalization. In both Cre$^-$ and Cre$^+$ mice, AAV-DIO-injected tibiae were normalized to the contralateral PBS-injected limb to account for unspecific differences in bone phenotype between animals, differences caused by CypD deletion among experimental and control mice, and unforeseen effects of virus infection. Normalized data showed a significant decrease in cortical thickness in the virus-infected Cre$^+$ mice (*Figure 7I*) and a decrease in tibial bone volume fraction in these mice which did not reach statistical significance (*Figure 7J*). These changes were sufficient to decrease tibial biomechanical properties and increase bone fragility in virus-injected bones from Cre$^+$ mice

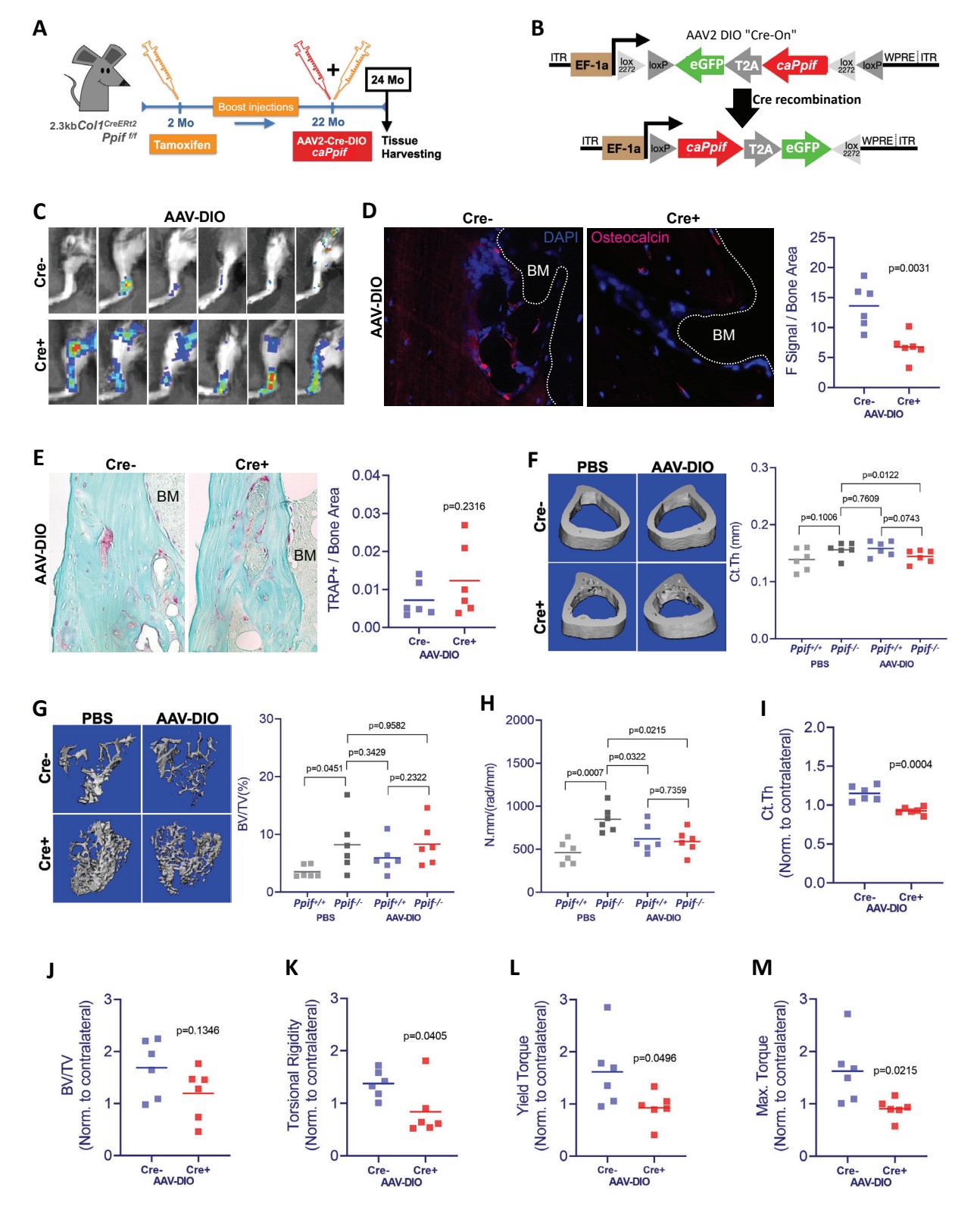

**Figure 7.** Cyclophilin D (CypD) re-expression and gain-of-function in vivo decreases bone mechanical properties. (**A**) Experimental design: CypD deletion was induced in 2-month-old OB-specific inducible (2.3 kb *Col1^CreERt2*;*Ppif^f/f*) CypD loss-of-function (LOF) mice. Virus intra-bone marrow injection performed in the right tibia at 22 months. (**B**) Diagram showing the AAV2-Cre-DIO-*caPpif*-eGFP construct. The gene of interest is inserted in the vector in antisense orientation and is flanked by double floxed sites. In cells expressing Cre recombinase, the gene of interest and eGFP reporter are flipped

*Figure 7 continued on next page*

*Figure 7 continued*

and 'turned-on'. (**C**) eGFP signal captured by IVIS in vivo confirming the successful viral infection and recombination in the tibia of Cre$^+$ mice 2 months after intra-bone marrow injection; CypD re-expression decreased osteocalcin immunofluorescence signal (**D**) but it did not change osteoclast activity measured by tartrate-resistant acid phosphatase (TRAP) staining (**E**) (BM: bone marrow). Bone volumes and biomechanical properties of CypD deletion and CypD re-expressing bones were measured by micro-computed tomography (µCT) and a torsion test, respectively. CypD conditional deletion did not affect cortical thickness (**F**) however, it showed protective effect against trabecular bone volume loss in aging (**G**) and improved torsional rigidity (**H**) (PBS: *Ppif$^{+/+}$* X *Ppif$^{-/-}$*). For analysis of bones with AAV-DIO-delivered CypD re-expression, data was normalized to the contralateral PBS-injected limb to account for differences in bone phenotype between animals. Cre$^+$ mice showed decreased bone volumetric parameters and mechanical properties when compared to Cre$^-$ mice. (**I**) Cortical thickness (Ct Th). (**J**) Bone over total volume (BV/TV). (**K**) Torsional rigidity. (**L**) Yield torque. (**M**) Maximum torque. Plots show the actual data points from six independent mice per group, calculated means, and p value determined by an unpaired t-test. Paired t-test was used when left and right tibia from the same mouse were compared. Specimens' genotype guide: $^{PBS}$-*Ppif$^{+/+}$*: wild type; $^{PBS}$-*Ppif$^{-/-}$*: CypD conditional KO; $^{AAV-DIO}$-*Ppif$^{+/+}$*: wild type caCypD rescue; $^{AAV-DIO}$-*Ppif$^{-/-}$*: CypD conditional KO caCypD rescue.

The online version of this article includes the following figure supplement(s) for figure 7:

**Figure supplement 1.** Physiological role of *PPIF* transcriptional regulation controlled by BMP/Smad signaling.

**Figure supplement 2.** Bone volumetric parameters and biomechanical properties in mice with osteoblast-specific cyclophilin D (CypD) conditional deletion with or without CypD re-expression.

**Figure supplement 3.** Summary of the effects of *Ppif* manipulation in different mouse models from previous and current publication.

when compared to Cre$^-$ mice (***Figure 7K–M***). Since bone homeostasis is achieved by the interplay of distinct tissues and cell populations such as BMSCs, OBs, osteoclasts, myeloid, lymphoid, endothelial, sensory, and myogenic cells, as expected, some bone morphological parameters are not affected by CypD/MPTP manipulation targeted in OBs (***Figure 7—figure supplement 2B-S***). Nonetheless, our aggregated data present strong evidence that OB-specific CypD re-expression and GOF detrimentally affect OB function and it is sufficient to impair bone phenotype in aging.

## Discussion

Cellular differentiation usually challenges cells with significantly increased energetic demands when compared to the undifferentiated state. Fully differentiated cardiomyocytes and neuronal cells are characterized by a higher demand of energy and therefore an elevated mitochondrial activity when compared to their undifferentiated pairs. Such energetic shift requires OxPhos activation which relies on a higher $\Delta\phi$ m and proton motive force, fueled by increased electron flow in the electron transport chain. There is mounting evidence that an important part of the above-mentioned bioenergetic switch is a closure of the MPTP. As cells become more active, OxPhos activation leads to increases in ROS, byproducts of active respiration. ROS can then trigger MPTP opening, leading to loss of integrity of the IMM and of OxPhos function and ultimately prompting mitochondrial dysfunction, inflammation, and catabolic pathways (***Fayaz et al., 2015***; ***Carraro and Bernardi, 2016***). Although the energetic state of mitochondria can be regulated through several pathways, CypD is an important indirect regulator of OxPhos and the key opener of MPTP (***Alavian et al., 2014***; ***Beutner et al., 2017***; ***Porter and Beutner, 2018***). MPTP closure is beneficial for OxPhos activity, desensitizing mitochondria to higher ROS levels, and preserving mitochondrial membrane integrity.

CypD is regarded as the master regulator of MPTP, and a range of post-translation modifications have been described to exert regulatory properties on CypD activity and, consequently, on MPTP opening. However, how CypD expression is transcriptionally regulated has never been described. We found through an in silico analysis that *PPIF* promoter has multiple SBEs in the 1.1 kb 5' upstream region of both human *PPIF* and mouse *Ppif* gene. Smads are TFs downstream of the BMP signaling pathway and since OB differentiation is controlled at least in part via BMP/Smad signaling, we hypothesized that in the case of *Ppif*, Smads are inhibitory TFs. BMPs are a group of cytokines belonging to the transforming growth factor-β superfamily of proteins, which among other functions, induce stem cell differentiation. The BMP canonical pathway is Smad dependent, which upon phosphorylation and Smad4 binding, forms a heterodimeric complex. The complex translocates into the nucleus and activates or represses expression of a variety of genes (***Wang et al., 2014***; ***Nickel and Mueller, 2019***). As expected, our DNA-protein binding ChIP analysis revealed stronger Smad1-*Ppif* interaction when ST2 cells were supplemented with BMP2 for 24 hr compared to vehicle control-treated cells. Although Smad1-*Ppif* interaction was found to be present at lower levels in the non-BMP-treated ST2 cells, this

finding is consistent with the fact that intrinsic levels of BMP activity are present in undifferentiated osteogenic cells. Since we have shown that pCMV-Smad1 transfection in ST2 and MC3T3e1 cells downregulated CypD mRNA expression, our results demonstrate that increased Smad1 interaction with CypD gene (*Ppif*) promoter leads to *Ppif* transcriptional repression. The specificity and functionality of BMP/Smad signaling in *Ppif* transcriptional repression was further confirmed using a *Ppif*-luc reporter with point mutations in the SBEs and delineated using the functional 1.1 kb *Ppif*-luc reporter with or without BMP/Smad inhibitors. However, the crosstalk between BMP and Wnt signaling could indirectly repress *Ppif* activity by other means than just Smad1-*Ppif* binding. Our in silico analysis revealed a potential TCF/LEF1 binding site within the *Ppif* promoter which was shown to have no effect on *Ppif* regulation. However, Runx2 is a downstream target of BMP/Smad signaling known to regulate several genes responsible for growth control and differentiation. Other TFs downstream Runx2 activation could also directly/indirectly exert regulatory function on *Ppif* transcription. Therefore, we investigated the role of BMP/Smad signaling in CypD downregulation and *Ppif* repression in 143b-OS cells, where osteogenic signaling downstream of BMP/Smad is arrested. Confirmatory results for 143b-OS cells strongly indicate that BMP/Smad signaling is in fact a direct transcriptional repressor of *PPIF* and is sufficient to repress *PPIF* without any additional downstream differentiation signals.

To our knowledge, this is the first evidence for TF-mediated regulation of *PPIF* transcription. Although our main focus was on osteogenic cells, our findings can be potentially extrapolated and studied in other cell types. 143b-OS cells' results indicate that Smad-mediated CypD downregulation/*Ppif* repression is not exclusive to osteogenic differentiation. As previously discussed, BMP is ubiquitously expressed and described to induce cellular differentiation and maturation in various tissues. Accordingly, some cells induced by BMP/Smad signaling reprogram their metabolic profile to a higher energetic state during differentiation and are shown to decrease CypD mRNA expression and/or activity, such as cardiomyocytes and neuronal cells. We previously reported that BMP2 induction stimulates mitochondrial OxPhos during OB differentiation and that such activation is at least partially driven by a metabolic signaling (*Smith and Eliseev, 2021*). Myocardial differentiation from cardiac progenitors is regulated by BMP/Smad signaling. BMP2/4 conditional KO impairs myocardial differentiation by drastically reducing sarcomeric myosin (*Wang et al., 2014*). In neuronal cells, BMP signaling is pivotal for cellular fate specification in both neurogenesis and astrogliogenesis by regulating transcriptional activity (*Bond et al., 2012*). Several BMP ligands, including BMP2, are described to regulate central nervous system development and patterning. Therefore, it is logical to suggest that CypD downregulation mediated by BMP-dependent, R-Smad (Smad1, -5, -8) transcriptional repression of *Ppif* gene can be present in other tissues during development and maintenance. However, we found that mouse myogenic C2C12 cells upregulate *Ppif* activity when Smad-dependent BMP signaling is activated, suggesting that the effect direction of Smads on *PPIF* regulation is tissue-specific. This bi-directional effect of Smads on gene regulation is well described. Smads can either act as promoter repressors or activators in a gene-specific manner (*Massagué et al., 2005*). The tissue-specific effect of Smad1 on *Ppif* can be explained, for instance, by the fact that cell/tissue specification is driven by selective methylation of promoter regions which changes chromatin accessibility (*Atlasi and Stunnenberg, 2017*). Such modification can either completely repress promoter activity or alter its regulatory mechanism. We showed that *Ppif* promoter bashing using different promoter regions (P, M+P, or D+M) led to promoter activation instead of repression in osteogenic ST2 cells overexpressing Smad1. This effect is similar to the effect observed in myogenic C2C12 cells using the full-length *Ppif* reporter. It remains to be determined which SBE combinations are accessible in C2C12 cells and how it would impact the recruitment of Smads and co-factors. The role of CypD expression is not well studied in myogenic program but RNAseq studies revealed no changes in CypD expression during myoblast differentiation (*Zeng et al., 2016*). Interestingly, BMP signaling is suppressed during myogenic differentiation showing marked upregulation of Noggin expression (*Ono et al., 2011*). In fact, Noggin or Wnt3a administration stimulates myogenic differentiation presenting synergic effect when combined (*Shang et al., 2007*; *Terada et al., 2013*). Conversely, addition of BMP ligands to C2C12 cells is shown to suppress myogenic differentiation but BMPs are necessary for muscle hypertrophy (*Borok et al., 2020*). Clearly, a fine-tuning of BMP regulation is required in muscle tissue. Since Smad1 transfection increased *Ppif* activity in myogenic cells, we can speculate that constitutive activation of R-Smads is more associated with inflammatory and catabolic

pathways in muscle tissue. Not surprisingly, BMP hyperactivation is described to be associated with impaired muscle regeneration (*Borok et al., 2020*).

The effects of CypD on cell functioning have been studied for over 35 years. First, CypD was exclusively attributed to regulate cell death, however later studies provided further understanding of CypD function within mitochondria and its importance for cell physiology and bioenergetic response. The CypD KO mouse model has been vastly applied to study the effect of MPTP downregulation in vivo. CypD deletion is shown to present a protective phenotype against aging in several tissue-specific mouse models. Additionally, CypD ablation in mice also resulted in diminished disease progression for some pathologies, such as reperfusion injury of the heart and brain, axonopathy, and other neurodegenerative and demyelinating diseases (*Baines et al., 2005*; *Schinzel et al., 2005*; *Giorgio et al., 2010*; *Halestrap and Richardson, 2015*; *Ahier et al., 2018*; *Gauba et al., 2019*). In contrast, CypD overexpression or overactivation is related to the onset and progression of several pathological states, either during development or in aging. Cardiomyocytes unable of downregulating CypD during embryonic development can cause congenital heart defects, secondary to defects in MPTP activity and myocyte differentiation (*Hom et al., 2011*; *Lingan et al., 2017*). The progression of Alzheimer's disease is accelerated by higher CypD levels and MPTP activity, leading to further mitochondrial stress and neuronal exhaustion (*Alavian et al., 2014*; *Gauba et al., 2019*). Collagen VI myopathies are well described and established pathologies where CypD overexpression plays a major role, and as expected, treatment with CypD inhibitors can reverse the disease (*Giorgio et al., 2010*; *Zulian et al., 2014*).

In bone tissue, we have shown that BMSCs from CypD KO mice have higher osteogenic potential and OxPhos activity. Moreover, mice with CypD ablation presented with less osteoporosis burden, stronger bones in aging, and improved fracture healing (*Shum et al., 2016b*; *Shares et al., 2020*). OB function involves bone matrix deposition, an energy-consuming process, which according to our and others' data, is highly dependent on mitochondrial bioenergetics. In fact, our lab and others have demonstrated that BMSCs shift their bioenergetic profile toward OxPhos during OB differentiation (*Shum et al., 2016a*; *Feigenson et al., 2017*; *Colaianni et al., 2018*; *Shares et al., 2018*; *Yu et al., 2019*; *Smith and Eliseev, 2021*). Not surprisingly, we recently reported that higher OxPhos in BMSCs correlates with better spinal fusion outcomes in both human patients and in a mouse model (*Laura et al., 2020*). Taken together, the evidence collected in other groups' studies, our previous studies, and the present study points that OxPhos is activated while MPTP is inhibited via CypD downregulation during OB differentiation. Our data corroborate reports of CypD downregulation and MPTP closure during neuronal differentiation in developing rat brains, as well as during cardiomyocyte differentiation in developing mouse hearts (*Eliseev et al., 2007*; *Hom et al., 2011*; *Lingan et al., 2017*). However, the importance of CypD/MPTP downregulation for OB differentiation and bone maintenance was not totally clear. In this study, we used a CypD/MPTP GOF model that stably expressed caCypD K166Q mutant (*caPpif*) in either MC3T3-e1 cell line or in primary cells from a novel OB-specific CypD GOF mouse model. Acetylation of CypD at K166 position increases CypD activity and MPTP opening and can be mimicked by K166Q mutation (*Alkalaeva et al., 2009*; *Porter and Beutner, 2018*). The development of this mouse model allowed us to investigate the importance of CypD downregulation during osteogenic differentiation in primary mouse cells. Both MC3T3e1 cells and BMSCs showed decreased osteogenic markers, impaired OB differentiation and function at the time caCypD was expressed (from D0 in caCypD-transfected MC3T3e1 cells, and from D11 in osteoinduced BMSCs from CypD GOF mouse). In our in vivo model, CypD re-expression and GOF reversed the beneficial effects of CypD deletion on bone phenotype in aged mice. We have shown that CypD KO mice are protected against bone loss in aging, and present stronger bones when compared to wild type control littermate mice (*Shum et al., 2016b*). Additionally, we found that bones from aged mice present higher CypD levels when compared to young mice. CypD re-insertion in OBs in mice with OB-specific CypD deletion by tibial intra-bone marrow viral infection, delivering the *caPpif* transgene, rescued bone mechanical properties observed in aging. Our data suggests that OB-specific CypD/MPTP overactivation in vivo is related to a weaker bone phenotype as seen in advanced bone loss and osteoporosis. This result goes along with studies showing that CypD/MPTP upregulation is present in several tissues during aging and as discussed, such upregulation plays a role in the onset and progression of several degenerative pathologies. However, a limitation in our in vivo model is the fact that we have not analyzed vertebral bones, which are more relevant in the context of osteoporosis.

During de novo bone formation in fracture repair, we observed CypD downregulation coincident with upregulation in BMP2. This finding may provide explanation for our previously published report where we found no effects of CypD conditional deletion in *Prx1*-expressing osteoprogenitors. Since in young mice with functional BMP signaling, CypD is downregulated during fracture repair, the effect of any additional bone-specific CypD downregulation may be blunted. Delayed fracture repair is associated with aging when BMP signaling is low. Therefore, we can interconnect our previous and current data on *PPIF* regulation summarized in *Figure 7—figure supplement 3* and conclude that such regulation process is crucial for bone homeostasis and repair.

Overall, our findings establish BMP/Smad signaling as a transcriptional regulator of *PPIF* expression playing an important role in bone physiology and pathology. We provided evidence that CypD downregulation led by BMP signaling is important during osteogenic differentiation and that the increase in CypD expression in OBs during aging can be detrimental to bone phenotype and strength. Even though aged skeleton is characterized by a very low turnover rate, CypD overexpression for a total period of only 2 months was sufficient to decrease bone mechanical properties. These results highlight the importance of CypD regulation for a proper bone maintenance and positions CypD as a potential target for bone health. Additionally, RNAseq data from pathological ossification/calcification conditions, such as osteoarthritis and osteophyte formation (*Dunn et al., 2016*) and cardiac valve calcification (*Huang et al., 2019*), suggested the inverse correlation between BMP signaling and CypD expression in these pathologies. Therefore, the BMP/Smad-*Ppif* axis is also relevant for these pathological conditions and can be aimed as a potential therapeutic target as well. In fact, there are plenty of evidence showing that CypD can be therapeutically targeted to treat some pathological conditions. Studies in the heart and brain have shown that CypD inhibitors can protect against injury after reperfusion in animal models (*Halestrap and Richardson, 2015*; *Warne et al., 2016*). However, the lack of target validation and better understanding of CypD regulatory pathway has contributed to failed Phase III clinical trials (*Nighoghossian et al., 2015*; *Ong et al., 2017*; *Briston et al., 2019*). Therefore, our findings can be applied to develop new strategies to regulate CypD expression, and the similar strategy can be used to other fields of mitochondrial-mediated human pathologies characterized by CypD/MPTP deregulated activity.

## Materials and methods
### Mouse strains

*R26^caPpif* with C57Bl/6 genetic background, CypD GOF, mice were generated by the combined effort of our lab and Dr George Porter's lab in the University of Rochester Gene Targeting and Transgenic Core Facility. We used the CTV vector (CAG-STOP-eGFP-ROSA targeting vector, Addgene) described by Dr Changchun Xiao (a modification from the original vector in the Klaus Rajewsky lab) to knock in the transgene *caPpif* construct encoding constitutively active CypD K166Q mutant, in embryonic stem (ES) cells. In brief, *caPpif* cDNA containing Myc-DDT(Flag) tag on its N-terminal was cloned into the CTV vector containing NeoSTOP cassette flanked by loxP sites. This vector also contains IRES-eGFP for easy detection of a transgene expression. This vector was designed for CypD GOF while inserted at *Rosa26* locus. Transformed ES cells were injected into the blastocysts to produce chimeric mice followed by breeding the chimeras to germline transmitted offspring containing the knock-in *caPpif* gene. After the creation of first transgenic hemizygous mouse littermates *R26^caPpif/+*, the colony was expanded and later bred in house with OB-specific tamoxifen-inducible *Col1^CreERt2* mice, 2.3 kb variant (final cross is *Col1^CreERt2/+;R26^caPpif/+*). 2.3 kb *Col1^CreERt2* mice were a kind gift from Dr Ackert-Bicknell (formerly of University of Rochester, originated in the Karsenty Lab at Columbia University Department of Genetics). *Ppif* floxed (*Ppif^f/f*) mice were acquired from Jackson Laboratories (originated in the Korsmeyer lab at Dana Farber Cancer Institute, RRID: IMSR_JAX:005737) and bred in house with 2.3 kb *Col1^CreERt2* mice to allow OB-specific CypD deletion. C57BL/6J mouse strain was obtained from the Jackson Laboratory (RRID: IMSR_JAX:000664) and bred in house. All mice were housed at 23°C on a 12 hr light/dark cycle with free access to water and PicoLab Rodent Diet 20 (LabDiet #5053, St Louis, MO). Mice were in group housing when applicable based on weaning. Testing naïve mice with an average weight of 28 g were used for experiments. The assessments of animal studies were performed in a blinded and coded manner.

## Isolation of BMSCs

Primary bone marrow cells were harvested from femurs and tibiae bone marrow from $Col1^{CreERt2/+}$;$R$-$26^{caPpif/+}$ and Cre-negative control littermates or from C57BL/6J mice. Cells were plated at a density of $20 \times 10^6$ per 10 cm dish in physiological 'low' glucose (5 mM) DMEM (LG-DMEM) supplemented with 1 mM L-glutamine, 10% fetal bovine serum (FBS), 1% penicillin-streptomycin, and incubated at 37°C, 5% $CO_2$, and physiologically relevant at 5% $O_2$. BMSCs were selected by plastic adherence, and detached from plates using 0.25% trypsin/1 mM EDTA treatment, as previously described (*Shum et al., 2016a*).

## Mouse cell lines

Mouse long bone-derived ST2 cells were a gift from Dr Clifford Rosen (Maine Medical Center). C2C12 mouse myogenic cells were from ATCC and provided by Dr Calvin Cole (University of Rochester). Mouse calvarial bone-derived MC3T3-E1 cells were acquired from ATCC. Cells were expanded in sterile conditions and maintained in a 37°C incubator at 5% $CO_2$ in low glucose αMEM media (Gibco A10490-01) containing 1 mM L-glutamine, ribonucleosides (0.01 g/L each), deoxyribonucleosides (0.01 g/L each), no ascorbic acid, 10% FBS (Gibco 10437-028) heat inactivated for 30 min at 55°C, and 1% pennicillin/streptomycin (Gibco 15140-122). Cells were maintained at passage numbers <20 as recommended by the supplier and from previous handling experience. Cells have been authenticated by their osteogenic (ST2 and MC3T3-E1) or myogenic (C2C12) differentiation potential. They are mycoplasma-free.

## Human osteoblastic cell line

hFOB cells were purchased from ATCC and expanded in DMEM supplemented with 1 mM L-glutamine, 10% FBS, 1% penicillin-streptomycin, and incubated at 37°C and 5% $CO_2$. We used passage number between 3 and 7 for our experiments (*Shapovalov et al., 2010*; *Giang et al., 2013*).

## Human OS cells

143b human OS cells were acquired from ATCC (Manassas, VA). Cells were expanded in sterile conditions and maintained in a 37°C incubator at 5% $CO_2$ in DMEM supplemented with 1 mM L-glutamine, 10% FBS, and 1% penicillin-streptomycin.

## Osteoinduction

Cells were osteoinduced at confluency in their appropriate media either supplemented with 50 μg/mL ascorbate (TCI A2521) and 2.5 mM β-glycerolphosphate (USB Corp Cleveland, OH) for 7 and 14 days, or 50 ng/mL BMP2 (R&D Systems 355-BM-050/CF) for 5 days. OB differentiation was assessed by alkaline phosphatase (Thermo NBT/BCIP 1-step 34042) and alizarin red staining. Crystal violet stain (Sigma C3886) was used at 0.5% to determine total cell count as previously described (*Shares et al., 2018*).

## Real-time RT-qPCR

Total RNA was isolated using the RNeasy kit (Qiagen 74106) and reverse transcribed into cDNA using the qScript cDNA synthesis kit (Quanta 95048-500). cDNA was subjected to real-time RT-PCR. The primer pairs used for genes of interest are outlined in Key resources table. Real-time RT-PCR was performed in the RotorGene system (Qiagen) using SYBR Green (Quanta 95072-012). *Alp* and *Bglap* gene expression was used to confirm osteogenic differentiation.

## MPTP opening calcein-cobalt assay

To assess the MPTP activity, mitochondrial membrane integrity was measured using a method of calcein quenching by cobalt as described by Petronilli et al. in 1999. The assay is based on the fact that calcein accumulated in the cytosol is quenched after co-loading with cobalt, whereas calcein accumulated in mitochondria is not accessible to cobalt and therefore not quenched unless mitochondrial membranes are permeabilized. Cells were incubated with 1 μM calcein-AM (acetoxymethyl ester) in the presence of 1 mM cobalt chloride for 30 min at 37°C, washed, lifted from plates using a cell lifter, resuspended in PBS solution, and assayed using the BD Biosciences LSRII flow cytometer. Ionomycin at 1 μM was used as negative control in a set of calcein/cobalt-loaded cells to dissipate calcein signal.

## Mitochondrial mass assay

Cells were stained with nonyl acridine orange (NAO) (Invitrogen A1372) at 100 nM, a fluorescent probe that labels cardiolipin present primarily in mitochondrial membranes (*Beutner et al., 2014*), for 15 min at 37°C. Stained cells were then lifted from plates with cell lifter, washed, and resuspended in PBS. NAO signal was detected in BD Biosciences LSRII flow cytometer.

## Mitochondrial functional assay

$7 \times 10^4$ BMSCs from $R26^{caPpif/+}$ (CypD GOF) mice were plated on laminin-coated 25 mm round glass coverslips and infected with Ad5-CMV-Cre or Ad5-CMV-eGFP virus at 500 MOI. Forty-eight hrs after infection, BMSCs were induced in osteogenic media. At day 0 or day 11 of osteoinduction, cells were stained with the nuclear staining Hoechst 33343 (Molecular Probes H 1339) at 0.5 μM, and the $\Delta\phi$ m-dependent dye TMRE (Invitrogen T669) at 20 nM. FCCP (Abcam ab120081) at 0.5 μM was used as a negative control. Cells were imaged using inverted fluorescence microscope (Axioscope) and TMRE fluorescence signal was quantified using ImageJ. ΔF signal is given by the difference of TMRE signal measured immediately before and 30 s after FCCP addition.

## Western blot

Cells were lysed in a lysis buffer containing protease inhibitors and subjected to 4–12% sodium dodecyl sulfate polyacrylamide gel electrophoresis followed by transfer to polyvinylidene difluoride membranes and blocking in 5% dry milk reconstituted in PBST (PBS supplemented with Tween 20 at 0.05%), as previously described (*Shum et al., 2016a*). All antibodies were diluted in 2.5% dry milk in PBST. For CypD detection, blots were probed with monoclonal CypD antibody (RRID: AB_478283) diluted 1:1000 and HRP-conjugated goat anti-mouse antibody diluted 1:3000. To verify equal loading, blots were re-probed with either β-actin (RRID: AB_476697) or VDAC1 (RRID: AB_632587) antibody diluted 1:2000 and HRP-conjugated goat anti-mouse antibody diluted 1:5000. CypD and β-actin signals were developed with West Pico Substrate. Band densitometry was measured with Image Lab Software. CypD signal was normalized to β-actin.

## Tibial intra-bone marrow AAV2(serotype 2)-Cre-DIO-*caPpif* -eGFP infection

Transgene *caPpif* construct was cloned into the Double-floxed Inverted Orientation (DIO) Cre-On vector pAAV-Ef1a-DIO-eGFP-WPRE-pA (Addgene plasmid # 37084; http://www.addgene.org/37084/; RRID:Addgene_37084 – gift from Bernardo Sabatini – *Saunders et al., 2012*). DIO vector was cloned using the AscI and NheI restriction site, and transgene introduced through transgene-specific primers and PCR amplification. The AscI site was N-Terminal and the NheI site C-Terminal with respect to the transgene. Cloning and sequence confirmation was performed. Cloned vectors were amplified with recombination deficient bacteria (OneShot Stbl3, Invitrogen) and tested functionally for Cre-On expression by calcium phosphate transfection (Invitrogen) into HEK 293 cells constitutively expressing Cre. The AAV particles (AAV2-Cre-DIO-*caPpif*-eGFP) were produced at the University of North Carolina Viral Core Facility.

$Col1^{CreERt2/+};Ppif^{f/f}$ and Cre-negative $Ppif^{f/f}$ littermate mice were anesthetized using 100 mg/kg ketamine and 10 mg/kg xylazine IP at a rate of 0.1 mL per 10 g of body weight. Hair was shaved around the joint area and 70% alcohol and iodine were used to clean the area. A 1 mL syringe with a 26 G (3/8 length) needle was inserted into the intra-bone marrow tibial space by gentle twisting and pressuring at the top of the tibiae (proximal epiphyses) in the knee joint. The hole created by the 26 G needle was expanded by lateral and whirling movements. The 26 G needle was removed and a Hamilton syringe with a 22 S needle gauge (Z15364-8, 100 μL Hamilton, Gastight 1800, needle Gauge 22S) was inserted into the expanded hole created by the 26 G needle. Ten microliters of viral solution ($1.8 \times 10^{13}$ viral particles/mL AAV2-Cre-DIO-*caPpif* -eGFP) was slowly injected by free hand into the marrow space of the tibiae (injection technique and viral load adapted from [*Selenica et al., 2016*]). Contralateral tibiae were injected with 10 μL of sterile PBS solution as an intra-mouse control. Immediately after the viral injection, 100 μL of 3 mg/mL tamoxifen was intra-peritoneally injected 5 days in a row to induce Cre activation. eGFP signal confirming viral infection and subsequently OB recombination was captured by in vivo image system IVIS 30 and 60 days post-infection.

## In Vivo Imaging System

Under isoflurane anesthesia, lower limbs from virus-infected mice were shaved and imaged for eGFP signal using In Vivo Imaging System (IVIS) Spectrum (Caliper Life Sciences). A negative control mouse, which received only PBS intra-bone marrow injection on both tibiae, was used to subtract non-specific eGFP signal arising from bone autofluorescence.

Bone µCT following euthanasia, virus and PBS-injected tibiae were isolated and cleaned of excess soft tissue. Tibiae were stored at −80°C prior to µCT. Bones were imaged using high-resolution acquisition (10.5 µm voxel size) with the VivaCT 40 tomograph (Scanco Medical). Scanco analysis software was utilized for volume quantification.

Biomechanical torsion testing immediately following µCT scanning, tibiae were subjected to biomechanical testing. The ends of the tibiae were cemented (Bosworth Company) in aluminum tube holders and tested using an EnduraTec TestBench system (Bose Corporation, Eden Prairie, MN). The tibiae were tested in torsion until failure at a rate of 1°/s. The torque data were plotted against rotational deformation to determine maximum torque and torsional rigidity. Data from virus-injected tibiae were normalized to the contralateral PBS-injected (intra-mouse control) tibiae data.

## Histology

After biomechanical testing, tibiae bones were NBF-fixed and processed for histology via decalcification in Webb-Jee 14% EDTA solution for 1 week followed by paraffin embedding. Sections were cut to 5 µm in three levels of each sample, and then stained with either TRAP or IF.

## Immunofluorescence

NBF-fixed tibiae were processed as above. IF was carried out using a primary anti-osteocalcin antibody (RRID:AB_10540992) diluted 1:400 or anti-GFP antibody (RRID:AB_303395) diluted 1:500, followed by incubation with anti-rabbit IgG secondary antibody conjugated with Alexa Fluor647 (RRID:AB_2722623) diluted 1:2000 or with anti-rabbit IgG secondary antibody conjugated with Alexa Fluor488 (RRID:AB_2630356), respectively. Anti-BMP2 antibody (RRID:AB_2243574) and anti-CypD antibody (RRID:AB_10864110), both diluted 1:500, were incubated simultaneously, followed by incubation with anti-rabbit IgG secondary antibody conjugated with Alexa Fluor647 (RRID:AB_2722623) and anti-mouse IgG secondary antibody conjugated with Alexa Fluor488 (RRID:AB_2576208), both diluted 1:2000. The primary antibody solution was composed of PBS, 0.1% Tween-20%, and 5% Goat Serum. Prior incubation at 65°C in 10 mM sodium citrate (pH 6.0) for 3 hr was performed for antigen retrieval. Fluoroshield Mounting Medium with DAPI (ab104139) was used to counterstain and coverslip IF-slides.

## Histomorphometry

TRAP or IF-stained slides were scanned in an Olympus VSL20 whole slide imager at 40× magnification and evaluated with VisioPharm automated histomorphometry software. TRAP-stained slides were analyzed to measure the TRAP-positive area relative to total bone area in the tibiae shaft. IF-stained slides probed for osteocalcin were analyzed to measure the fluorescence signal intensity relative to total bone area of tibiae shaft. Three different levels were counted per mouse and averaged.

## ChIP assay

ChIP assay was performed using SimpleChIP Enzymatic Chromatin IP Kit – Magnetic Beads (Cell Signaling Technology, Inc, Danvers, MA) according to the manufacturer's instructions. Briefly, cells were fixed, and DNA cross-linked with formalin and homogenized. Nuclei were pelleted and chromatin was digested enzymatically, sonicated and immunoprecipitated with either anti-Smad1 antibody (RRID:AB_628261), or negative control immunoglobulin G, or positive control histone H3 antibody using protein G magnetic beads. Chromatin was eluted from immunoprecipitate complexes and cross-links reversed with NaCl and proteinase K. After purification, DNA was amplified by a PCR using primers to amplify the distal SBE region (−1107—947) within the *Ppif* promoter. Band density quantification for Smad1 immunoprecipitation in both conditions was adjusted by background subtraction and normalized to total DNA input. Unspecific signal from IgG band was also subtracted from Smad1-specific signal.

## Cloning of *Ppif* promoter into luciferase reporter and reporter activity assay

Mouse *Ppif* promoter fragments containing the –1.1 to –0.1 kb, or –0.37 to –0.1 kb, or –0.62 to –0.45 kb, or –1.1 to –0.95 kb, or –0.62 to –0.1 kb, or –1.1 to –0.45 kb region were PCR amplified from purified C3H/HeJ mouse DNA. Primers were designed to introduce CTCGAG XhoI 5′ and AAGCTT HindIII 3′ flanking sequences. For *Ppif* promoter mutagenesis, point mutations for each SBE found in the 1.1 kb *Ppif* promoter region were designed to substitute either a cytosine or guanine for an adenosine nucleotide in the sense strand, and therefore a complementary thymine base in the anti-sense strand. CTCGAG XhoI 5′ and AAGCTT HindIII 3′ flanking sequences were also introduced in the 1.1 kb mutated *Ppif* promoter, and a synthetic sequence fragment was ordered from IDT (IDT, gBlocks Gene Fragments). The fragments were then subcloned into the XhoI 3′/HindIII 5' site of the promoterless pGL4.10 luciferase reporter vector. The correct insert orientation of the resulting promoter reporters was verified by sequencing. To evaluate promoter activities, the constructed *Ppif*-Luc reporters were transfected into ST2, MC3T3e1, C2C12, or 143b-OS cells at 0.8 µg per well in 12-well plates. The promotorless renilla luciferase vector pRL (Promega, Madison, WI) was co-transfected at 50 ng per well as a reference. Smad1 activity was further activated with either 0.8 µg/mL pCMV-Smad1 co-transfection, 50 ng/mL BMP2, or osteogenic media induction. The role of BMP/Smad was delineated by co-transfecting inhibitory Smad7 using 0.8 µg/mL pCMV-Smad7 vector, or using BMP inhibitor, Noggin (R&D systems 1967-NG-025/CF) at 0.1 µg/mL. Wnt signaling role in *Ppif* promoter activity was assessed using Wnt3a (R&D systems 5036-WN-010) addition in the media. Firefly and renilla luciferase activities were measured using an Optocomp 1 luminometer and a Dual Luciferase Reporter Assay System (Promega) according to the manufacturer's protocol. The firefly luciferase signal was normalized to renilla luciferase signal and expressed as relative luminescence units fold change to pCMV empty vector control.

## Statistics

A power analysis on normalized biomechanical data was performed since it showed the highest variance. It was determined that some quantitative outcomes would require six mice per group. We set the significance level at 5% ($\alpha$=0.05) and Type II error ($\beta$) to ≤20%. For statistical analysis, we compared the difference of two simple groups independently, therefore an unpaired t-test was used when the frequency distribution of the differences between the two groups fitted a normal distribution. When left and right tibiae from the same animal was compared, we used a paired t-test. Although we analyzed independent variables and therefore independent hypothesis in the multi-group graphs (Ppif^f/f: Cre^+ or Cre^- × AAV-DIO or PBS), we performed ordinary one-way ANOVA using Dunnett's multiple comparisons test with a single pooled variance to further validate our statistical findings significance. Since no differences in significance were found, we maintained our t-test results.

## Acknowledgements

Funding was provided by the National Institute of Health grants R01 AR072601 to RAE, R21 AR070928 to RAE and JHJ, R01 AR070613 to HA, P30 AR069655 to the Center for Musculoskeletal Research, and Ruth L Kirschstein National Research Service Award from NIH Institutional Research Training Grant R90-DE022529-11 Training Program in Oral Science (TPOS) to RSJ.

---

## Additional information

### Funding

| Funder | Grant reference number | Author |
| --- | --- | --- |
| National Institute of Dental and Craniofacial Research | R90-DE022529 | Rubens Sautchuk |

---

| Funder | Grant reference number | Author |
|---|---|---|
| National Institute of Arthritis and Musculoskeletal and Skin Diseases | R21 AR070928 | Jennifer H Jonason Roman A Eliseev |
| National Institute of Arthritis and Musculoskeletal and Skin Diseases | R01 AR070613 | Hani A Awad |
| National Institute of Arthritis and Musculoskeletal and Skin Diseases | R01 AR072601 | Roman A Eliseev |
| National Institute of Arthritis and Musculoskeletal and Skin Diseases | P30 AR069655 | Rubens Sautchuk Brianna H Kalicharan Katherine Escalera-Rivera Jennifer H Jonason Hani A Awad Roman A Eliseev |

The funders had no role in study design, data collection and interpretation, or the decision to submit the work for publication.

## Author contributions

Rubens Sautchuk, Data curation, Formal analysis, Investigation, Methodology, Validation, Visualization, Writing – original draft, Writing – review and editing; Brianna H Kalicharan, Katherine Escalera-Rivera, Data curation, Formal analysis, Investigation, Methodology, Visualization; Jennifer H Jonason, Conceptualization, Data curation, Writing – original draft, Writing – review and editing; George A Porter, Conceptualization, Funding acquisition, Investigation, Methodology, Supervision, Writing – review and editing; Hani A Awad, Formal analysis, Funding acquisition, Investigation, Methodology, Supervision, Visualization, Writing – review and editing; Roman A Eliseev, Conceptualization, Data curation, Formal analysis, Funding acquisition, Investigation, Methodology, Project administration, Resources, Software, Supervision, Validation, Visualization, Writing – original draft, Writing – review and editing

## Author ORCIDs

Rubens Sautchuk http://orcid.org/0000-0002-0302-7562
Hani A Awad http://orcid.org/0000-0003-2197-2610
Roman A Eliseev http://orcid.org/0000-0002-6783-7388

## Ethics

This study was performed in strict accordance with the recommendations in the Guide for the Care and Use of Laboratory Animals of the National Institutes of Health. All of the animals were handled according to approved institutional animal care and use committee (IACUC) protocol (#2012-043) of the University of Rochester. All surgery was performed under anesthesia, and every effort was made to minimize suffering.

## Decision letter and Author response

Decision letter https://doi.org/10.7554/eLife.75023.sa1
Author response https://doi.org/10.7554/eLife.75023.sa2

# Additional files

## Supplementary files

• Transparent reporting form

## Data availability

All data generated or analysed during this study are included in the manuscript and supporting files.

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

# Appendix 1

## Appendix 1—key resources table

| Reagent type (species) or resource | Designation | Source or reference | Identifiers | Additional information |
|---|---|---|---|---|
| Gene (*Mus musculus*) | *Ppif* promoter (–1110…–1 bp) | NCBI | NM_005729.3 | |
| Gene (*Homo sapiens*) | *PPIF* promoter (–1110…–1 bp) | NCBI | NC_000080.6 | |
| Strain, strain background (*Mus musculus*, male) | C57BL6/J | Jackson Laboratories | RRID: IMSR_JAX:000664 | |
| Genetic reagent (*Mus musculus*, male) | *R26^caPpif* | This paper | | Global CypD GOF or *caPpif* mouse (C57Bl/6 background) |
| Genetic reagent (*Mus musculus*, male) | 2.3 kb *Col1^CreERt2^;R26^caPpif/+* | This paper | | Col1-specific CypD GOF or *caPpif* mouse (C57Bl/6 background) |
| Genetic reagent (*Mus musculus*, male) | (*Ppif^f/f*) Ppif ^tm1Mmos^/J, C57Bl/6 | Jackson Laboratories | RRID: IMSR_JAX:005737 | Originated in the Korsmeyer lab at Dana Farber Cancer Institute |
| Cell line (*Mus musculus*) | ST2 | DMSZ | RRID:CVCL_2205 | Gift from Dr Clifford Rosen (Maine Medical Center) |
| Cell line (*Mus musculus*) | MC3T3-E1 subclone 14 | ATCC | RRID:CVCL_5437 | |
| Cell line (*Mus musculus*) | C2C12 | ATCC | RRID:CVCL_0188 | Gift from Dr Calvin Cole (University of Rochester) |
| Cell line (*Homo sapiens*) | hFOB 1.19 | ATCC | RRID:CVCL_3708 | |
| Cell line (*Homo sapiens*) | 143b | ATCC | RRID:CVCL_2270 | |
| Transfected construct (*Mus musculus*) | pCMV6-*caPpif* | Origene | MR202183 | Plasmid construct to transfect and express caCypD |
| Transfected construct (*Enterobacteria phage P1*) | Ad5-CMV-Cre | Vector Biolabs | No:1045 | Viral particle construct to transfect and express Cre recombinase |
| Transfected construct (*Aequorea Victoria*) | Ad5-CMV-eGFP | Vector Biolabs | No:1060 | Viral particle construct to transfect and express eGFP |
| Transfected construct (*Mus musculus*) | AAV2-Cre-DIO-*caPpif*-eGFP | UNC GTC Vector Core | #AV8154 | Viral particle construct to transfect and express caCypD in vivo |
| Biological sample (*Mus musculus*) | Primary BMSCs from C57BL6/J mice | Jackson Laboratories | | Freshly isolated from *Mus musculus* |
| Biological sample (*Mus musculus*) | Primary BMSCs from *R26^caPpif/+* mice | This paper | | Freshly isolated from *Mus musculus* |
| Antibody | Anti-CypD (mouse monoclonal) a.k.a Cyclophilin F | Abcam | RRID: AB_10864110 | WB (1:1000) IF (1:400) |
| Antibody | Anti-β-actin (mouse monoclonal) | Abcam | RRID: AB_476697 | (1:2000) |
| Antibody | Anti-VDAC1 (mouse monoclonal) | Santa Cruz Biotechnology | RRID: AB_632587 | (1:2000) |
| Antibody | Goat anti-mouse (goat polyclonal, HRP conjugate) | Abcam | RRID: AB_478283 | (1:3000) to (1:5000) |
| Antibody | Anti-osteocalcin (rabbit polyclonal) | Enzo Life Sciences | RRID: AB_10540992 | (1:400) |
| Antibody | Anti-GFP (rabbit polyclonal) | Abcam | RRID: AB_303395 | (1:500) |
| Antibody | Anti-BMP2 (rabbit polyclonal) | Abcam | RRID: AB_2243574 | (1:500) |
| Antibody | Goat anti-rabbit IgG (goat polyclonal, Alexa Fluor647 conjugate) | Abcam | RRID: AB_2722623 | (1:2000) |
| Antibody | Goat anti-rabbit IgG (goat polyclonal, Alexa Fluor488 conjugate) | Abcam | RRID: AB_2630356 | (1:2000) |
| Antibody | Anti-Smad1 (mouse monoclonal) | Santa Cruz Biotechnology | RRID: AB_628261 | (1:2000) |

*Appendix 1 Continued on next page*

*Appendix 1 Continued*

| Reagent type (species) or resource | Designation | Source or reference | Identifiers | Additional information |
|---|---|---|---|---|
| Recombinant DNA reagent | pCMV5-Smad1 (plasmid) | Addgene | #14045 | |
| Recombinant DNA reagent | pCMV5-Smad7 (plasmid) | Addgene | #11733 | |
| Recombinant DNA reagent | pCMV6-empty backbone (plasmid) | Origene | PS100001 | |
| Recombinant DNA reagent | pGL4.10 (plasmid) | Promega | PR-E6651 | |
| Recombinant DNA reagent | pRL (plasmid) | Promega | PR-E2241 | |
| Recombinant DNA reagent | XhoI (restriction enzyme) | New England BioLabs | R0146S | |
| Recombinant DNA reagent | HindIII (DNA ligase) | New England BioLabs | R3104S | |
| Recombinant DNA reagent | T4 (restriction enzyme) | New England BioLabs | M0202S | |
| Recombinant DNA reagent | CAG-STOP-eGFP-ROSA26TV (plasmid) | Addgene | # 15912 | Plasmid construct for gene knock-in engineering |
| Recombinant DNA reagent | *caPpif* | **Bochaton et al., 2015** | | caCypD construct (CypD GOF) |
| Recombinant DNA reagent | pAAV-Ef1a-DIO-eGFP-WPRE-pA (plasmid) | Addgene | # 37084 | Plasmid construct to create recombinant AAV |
| Sequence-based reagent | *B2m*_Fwd | IDT | RT q-PCR primer | AATGGGAAGCCGAACATAC |
| Sequence-based reagent | *B2m*_Rev | IDT | RT q-PCR primer | CCATACTGGCATGCTTAACT |
| Sequence-based reagent | *Smad1*_Fwd | IDT | RT q-PCR primer | GACAAGTTTATTTTCCTTTACAGGTCC |
| Sequence-based reagent | *Smad1*_Rev | IDT | RT q-PCR primer | CCACACACGGCAGTAAATG |
| Sequence-based reagent | *Runx2*_Fwd | IDT | RT q-PCR primer | CCGGGAATGATGAGAACTAC |
| Sequence-based reagent | *Runx2*_Rev | IDT | RT q-PCR primer | CCGTCCACTGTCACTTTAATA |
| Sequence-based reagent | *Ppif*_Fwd | IDT | RT q-PCR primer | CATGTACCC GAACAGAAC |
| Sequence-based reagent | *Ppif*_Rev | IDT | RT q-PCR primer | CATGTACCC GAACAGAAC |
| Sequence-based reagent | *Alp*_Fwd | IDT | RT q-PCR primer | CATGTACCC GAACAGAAC |
| Sequence-based reagent | *Alp*_Rev | IDT | RT q-PCR primer | GGGCTCAAA GAGACCTAAGA |
| Sequence-based reagent | *Bglap*_Fwd | IDT | RT q-PCR primer | GACCTCACAGATGCCAAG |
| Sequence-based reagent | *Bglap*_Rev | IDT | RT q-PCR primer | CAAGCCATACTGGTCTGATAG |
| Sequence-based reagent | *GAPDH*_Fwd | IDT | RT q-PCR primer | GAGTCAACGGATTTGGTCGT |
| Sequence-based reagent | *GAPDH*_Rev | IDT | RT q-PCR primer | GACAAGCTTCCCGTTCTCAG |
| Sequence-based reagent | *RUNX2*_Fwd | IDT | RT q-PCR primer | TCCGGAATGCCTCTGCTGTTATGA |
| Sequence-based reagent | *RUNX2*_Rev | IDT | RT q-PCR primer | ACTGAGGCGGTCAGAGAACAAACT |
| Sequence-based reagent | *ALP*_Fwd | IDT | RT q-PCR primer | TGCAGTACGAGCTGAACAGGAACA |
| Sequence-based reagent | *ALP*_Rev | IDT | RT q-PCR primer | TCCACCAAATGTGAAGACGTGGGA |
| Sequence-based reagent | *IBSP*_Fwd | IDT | RT q-PCR primer | AACGAACAAGGCATAAACGGCACC |
| Sequence-based reagent | *IBSP*_Rev | IDT | RT q-PCR primer | CCCACCATTTGGAGAGGTTGTTGT |
| Sequence-based reagent | *BGLAP*_Fwd | IDT | RT q-PCR primer | CCCTCACACTCCTCGCCCTATT |
| Sequence-based reagent | *BGLAP*_Rev | IDT | RT q-PCR primer | ATAGGCCTCCTGAAAGCCGATGT |
| Sequence-based reagent | *SMAD1*_Fwd | IDT | RT q-PCR primer | GACAAGTTTATTTTCCTTTACCCGTCC |
| Sequence-based reagent | *SMAD1*_Rev | IDT | RT q-PCR primer | CCACACAGGGCAGTAAAT |
| Sequence-based reagent | *PPIF*_Fwd | IDT | RT q-PCR primer | GCCGCAACACAAACGGTTCTC |

*Appendix 1 Continued on next page*

*Appendix 1 Continued*

| Reagent type (species) or resource | Designation | Source or reference | Identifiers | Additional information |
|---|---|---|---|---|
| Sequence-based reagent | *PPIF*_Rev | IDT | RT q-PCR primer | CCCGTCATCTCCTTCCTTCAATTCTC |
| Sequence-based reagent | Distal SBE_Fwd (CTCGAG XhoI 5′ flanking sequence) | This paper (IDT) | PCR cloning primer | AAGACTCGAGTGGAGATTCCCCGCTAT |
| Sequence-based reagent | Distal SBE_Rev (AAGCTT HindIII 3′ flanking sequence) | This paper (IDT) | PCR cloning primer | GGTAATTTCTCATCGCTTCCTTGAAGCTTAAGA |
| Sequence-based reagent | Middle SBE_Fwd (CTCGAG XhoI 5′ flanking sequence) | This paper (IDT) | PCR cloning primer | AAGACTCGAGATTCCAGGGGGTGTAAATCTA |
| Sequence-based reagent | Middle SBE_Rev (AAGCTT HindIII 3′ flanking sequence) | This paper (IDT) | PCR cloning primer | AGGATCTGGTCTCTAGAAGCAAAAAAGCTTAAGA |
| Sequence-based reagent | Proximal SBE_Fwd (CTCGAG XhoI 5′ flanking sequence) | This paper (IDT) | PCR cloning primer | AAGACTCGAGTTCTGTTATCTCTCCCTTTCTG |
| Sequence-based reagent | Proximal SBE_Rev (AAGCTT HindIII 3′ flanking sequence) | This paper (IDT) | PCR cloning primer | AAGCCAGCCGACCAATAAAAAGCTTAAGA |
| Sequence-based reagent | Cre_Fwd | IDT | PCR primer | CCTGGAAAATGCTTCTGTCCGTTTGCC |
| Sequence-based reagent | Cre_Rev | IDT | PCR primer | GAGTTGATAGCTGGCTGGTGGCAGAT |
| Sequence-based reagent | caCypD (TetIRES)_Fwd | IDT | PCR primer | AATGGCTCTCCTCAAGCG |
| Sequence-based reagent | caCypD (TetGFP)_Rev | IDT | PCR primer | GCGGATCTTGAAGTTCACCTTGATGCCGT |
| Peptide, recombinant protein | BMP2 | R&D Systems | 355-BM-050/CF | |
| Peptide, recombinant protein | Wnt3a | R&D Systems | 5036-WN-010 | |
| Peptide, recombinant protein | Noggin | R&D Systems | 1967 NG-025/CF | |
| Commercial assay or kit | SimpleChIP Enzymatic Chromatin IP Kit (Magnetic Beads) | Cell Signaling Technology | #9003 | |
| Commercial assay or kit | Dual Luciferase Reporter Assay System | Promega | E1960 | |
| Chemical compound, drug | L-Ascorbic Acid 2-Phosphate Sesquimagnesium Salt | TCI America | TCI A2521 | |
| Chemical compound, drug | Alkaline phosphatase substrate | Thermo Fisher | 34042 | 1-Step NBT/BCIP Substrate Solution |
| Chemical compound, drug | Calcein-AM | BD Biosciences | 354216 | |
| Chemical compound, drug | Cobalt chloride | Sigma-Aldrich | 232696 | |
| Chemical compound, drug | Ionomycin | Invitrogen | 124222 | |
| Chemical compound, drug | NAO | Invitrogen | A1372 | |
| Chemical compound, drug | Hoechst 33343 | Molecular Probes | H 1339 | |
| Chemical compound, drug | TMRE | Invitrogen | T669 | |
| Chemical compound, drug | FCCP | Abcam | Ab120081 | |
| Chemical compound, drug | Tamoxifen | Sigma-Aldrich | T5648 | |
| Chemical compound, drug | 4-OH-tamoxifen | Sigma-Aldrich | H7904 | |

