## [Editor Report]

This study provides evidence of cyclophilin-D transcriptional regulation of osteoblast differentiation and offers new insights into the underlying mechanisms identifying BMP/Smad signaling as a major mediator. The study has significant relevance and implication to bone health and offers a strong foundation for future work to decipher its unique role in different cells of the mesenchymal lineage and relevance to various bone-related diseases.

---

## [Decision Letter]

**Decision letter after peer review:**

Thank you for submitting your article "Transcriptional regulation of cyclophilin D by BMP/SMAD signaling and its role in osteogenic differentiation" for consideration by *eLife*. Your article has been reviewed by 3 peer reviewers, one of whom is a member of our Board of Reviewing Editors, and the evaluation has been overseen by Mone Zaidi as the Senior Editor. The following individual involved in review of your submission has agreed to reveal their identity: Narayan Avadhani (Reviewer #3).

Essential revisions:

1) Other pathways are important regulators of bone formation. It would be interesting to know if cyclophilin D is regulated by Wnt signaling, or if the regulation is BMP exclusive. Additionally, do the Ppif knock down or use of Smad inhibitor affect only the osteoblastic pathway or other skeletal-muscular pathways?

2) Previous published work by the authors demonstrated that loss of cyclophilin D on Prx1+ cells did not have an effect on bone cell function and on bone mass in the limb skeleton while global deletion of cyclophilin D had strong negative consequences on bone mass and fracture healing. These findings are not easily compatible with the authors contention that BMP signaling in bone forming cells regulates mitochondrial activity, as most if not all bone forming cells arise from Prx1+ progenitors. It would be helpful if the authors could provide a stronger explanation for these disparities.

3) Figure 2: rigor and presentation of data are marginal. Specificity of the binding should be tested using SBE-mutant vectors. Data should be normalized to input in both vehicle and BMP2 treated conditions.

4) Figure 3: Although the Luc reporter assay is important to demonstrate functionality, an appropriate negative control (mutant) derived from the 1.1 kb Ppif construct (does not bind Ppif) is needed.

5) Figure 4: The data describing Mt integrity are not very informative and should be supplemented with mitochondrial function assays.

*Reviewer #1 (Recommendations for the authors):*

General Comments:

1. The study focuses on BMP signaling and dismisses other osteogenic pathways such as WNT and β-catenin signaling. It would be very informative and would significantly enhance the study if the authors determine if the CypD mechanism is BMP-specific or not.

2. Osteogenesis and mitochondrial activity vastly change during bone diseases linked to obesity, aging, etc. It would be very exciting to determine the role of CypD in a disease/aging setting to highlight the relevance of the proposed mechanism and its potential utility as a therapeutic tool. This would constitute a major advance.

Specific comments:

3. Introduction: there are several redundant statements that hinder flow and should be streamlined.

4. The hypothesis is scantily articulated. All cells undergo OxPhos bioenergetic shift to facilitate differentiation. In fact, a generic argument can be made for other pathways that regulate OB differentiation. Nevertheless, the study does very little to address the hypothesis that "MTPTP inhibition via CypD transcriptional downregulation is required to sustain OxPhos function". The concept requires revision and better articulation

Results:

5. Figure 2: rigor and presentation of data are marginal. Specificity of the binding should be tested using SBE-mutant vectors. Data should be normalized to input in both vehicle and BMP2 treated conditions.

6. Figure 3: Although the Luc reporter assay is important to demonstrate functionality, an appropriate negative control (mutant) derived from the 1.1 kb Ppif construct (does not bind Ppif) is needed.

7. Figure 4: The data describing Mt integrity are not very informative and should be supplemented with mitochondrial function assays.

8. Figure 6: the CypD GOF experiments (caCypD) corroborate previous findings in terms of osteogenesis and as such offer marginal information. At minimum, OxPhos and other Mt functional properties should be characterized in the GOF model. Furthermore, cyclosporin A is a well-known CypD inhibitor which can be used to validate this system. Despite these confirmatory findings, it remains unclear how this information advances current knowledge, which is judged as incremental (please refer to general comments).

In sum, the study requires major revision including additional experiments avoiding confirmatory data. The general role of CypD in osteogenesis is already known, yet its utility in pathologic bone loss and additional insights in its transcriptional regulation if developed further, to address novel aspects, would fill important knowledge gaps and elevate significance of this research.

*Reviewer #2 (Recommendations for the authors):*

General comments: The authors propose a novel hypothesis about the regulation of cell function by BMP signaling and here use bone cells as a model system to test this hypothesis. While data obtained from individual experiments are convincing, overall, several conclusions require additional data or clarification and there are some missed opportunities to strengthen the manuscript.

Specific comments:

1. Previous published work by the authors demonstrated that loss of cyclophilin D on Prx1+ cells did not have an effect on bone cell function and on bone mass in the limb skeleton while global deletion of cyclophilin D had strong negative consequences on bone mass and fracture healing. These findings are not easily compatible with the authors contention that BMP signaling in bone forming cells regulates mitochondrial activity, as most if not all bone forming cells arise from Prx1+ progenitors. It would be helpful if the authors could provide a stronger explanation for these disparities.

2. BMP activity in bone is known to decline with age, consistent with the idea that reduced BMP signaling may be a cause of age-related bone loss. The authors have an opportunity to directly examine BMP signaling in the aging skeleton and determine if reduced signaling correlated with mitochondrial pore opening and loss of bone cell function.

3. The disease FOP is a gain of function of BMP signaling. The authors might consider examining the mouse model of FOP to see if there is a change in the mitochondrial physiology/pore opening in this disease.

4. Other pathways are important regulators of bone formation. It would be interesting to know if cyclophilin D is regulated by Wnt signaling, or if the regulation is BMP exclusive.

*Reviewer #3 (Recommendations for the authors):*

This is a relatively well done study which uses promoter analysis for transcription factor (Smad) binding using mutational analysis of promoter-reporter constructs and ChIP analysis. They use mouse ST2 and MC3T3-e1 cells for in vitro studies and confirm most of these in vitro findings using CypD null mice. In general, this manuscript should be acceptable for publication in *eLife*.

The main concern is that the studies may not address the wide enormity of this pathway with multiple BMPs and Smads affecting an array of physiological process. For example, do the Ppif knock down or use of Smad inhibitor affect only the osteoblastic pathway or other skeletal-muscular pathways? The second point is that the transcription pathway is active in a basal level suggesting that there may be some dose-effect in the modulation of different physiological pathway. The authors do not address this issue.

Most of the reagents are commercially available and this is not an issue.

[Editors’ note: further revisions were suggested prior to acceptance, as described below.]

Thank you for resubmitting your work entitled "Transcriptional regulation of cyclophilin D by BMP/SMAD signaling and its role in osteogenic differentiation" for further consideration by *eLife*. Your revised article has been evaluated by Mone Zaidi (Senior Editor) and a Reviewing Editor.

The manuscript has been modestly improved but there are some important remaining issues that need to be addressed, as outlined below. These concerns must be addressed experimentally before the manuscript can be considered further.

*Reviewer #2 (Recommendations for the authors):*

The authors have not satisfactorily addressed my major concerns, reiterated below:

Prior publications by the authors reported that global deletion of cypD resulted in no obvious skeletal phenotype until mice began to age when wt mice lost bone mass and cypD null mice did not undergo age related bone loss. Next they reported that global loss of cypD enhanced fracture repair in older mice. They later published that loss of cypD in Prx1+ cells did not positively affect bone function stating "lineage deletion of cypD is not sufficient to improve fracture repair," and that it was the systemic loss of cypD that affected the skelton. These data raise 3 issues:

a) In mice, Prx1+ cells make up the majority of cells forming the fracture callus and new bone, making it difficult to understand what osteolineage-specific effects are regulated by cypD.

b) Fracture repair cannot occur in the absence of BMP signaling in Prx1+ cells; if BMP signaling is an important physiological regulator of cypD in osteolineage cells, there should be a relationship between the loss of BMP signaling and loss of cypD in this population.

c) Without addressing the in vivo models suggested (FOP, aging,) and demonstrating that changes in BMP signaling in vivo in a physiologically relevant model result in changes in cypD in osteolineage cells, the study does not reach a level of physiological significance.

Since the authors have utilized a variety of mouse models and have not reported the complete phenotype of any, I suggest including a table that clearly delineates and compares the specific skeletal manifestations of cypD genetic deletion, cypD pharmacological deletion, and lineage-specific cypD deletion. Bone mass, bone strength and ability to repair fractures should be included as well as levels of BMP signaling in the skeletal tissues that demonstrate a phenotype.

---

## [Author Response]

Essential revisions:1) Other pathways are important regulators of bone formation. It would be interesting to know if cyclophilin D is regulated by Wnt signaling, or if the regulation is BMP exclusive. Additionally, do the Ppif knock down or use of Smad inhibitor affect only the osteoblastic pathway or other skeletal-muscular pathways?

We have added experiments using the Wnt signaling activator Wnt3a (Supplementary Figure S4) and C2C12 myogenic cells (Supplementary Figure S1E, F), and further discussed the results in the Discussion section.

2) Previous published work by the authors demonstrated that loss of cyclophilin D on Prx1+ cells did not have an effect on bone cell function and on bone mass in the limb skeleton while global deletion of cyclophilin D had strong negative consequences on bone mass and fracture healing. These findings are not easily compatible with the authors contention that BMP signaling in bone forming cells regulates mitochondrial activity, as most if not all bone forming cells arise from Prx1+ progenitors. It would be helpful if the authors could provide a stronger explanation for these disparities.

Global CypD deletion actually led to positive effects on bone phenotype in either aging^(1)^ and bone fracture healing^(2)^. Accordingly, we demonstrated in Figure 7 that Col1-conditional CypD deletion is also protective against bone loss and fragility in aging. Both models, previous and current, studied CypD deletion effects in aging focusing on bone maintenance and have no correlation to a bone repair model, a totally different process. Prx1-conditional CypD deletion showed no effects in bone fracture healing^(2)^ which is a model beyond the scope of this manuscript. Fracture repair involves several other signaling pathways and active function of several other cell types, as example of a vast list of cells from hematopoietic origin and mostly important, the transition of Prx1 progenitors cells to chondrocytes^(3)^ and chondrocyte trans-differentiation into osteoblasts, which do not occur in the process of bone maintenance.

3) Figure 2: rigor and presentation of data are marginal. Specificity of the binding should be tested using SBE-mutant vectors. Data should be normalized to input in both vehicle and BMP2 treated conditions.

ChIP assay is meant to characterize DNA-protein interactions in the native chromatin^(4, 5)^. Binding specificity in a ChIP assay is controlled by using properly washing protocol to remove non-specifically bound chromatin, negative control such as IgG antibody, and the PCR reaction carried by primers designed to flank a stretch of the region of interest^(4, 5)^. We agree that specificity must be further checked, so we addressed this issue using mutant vectors in the Luc-reporter assay (Item #4). ChIP data (Figure 2C) were in fact normalized to total DNA input as previously described in the figure legend. We briefly rearranged the legend text to make it clearer and also included the normalization info in the methods section.

4) Figure 3: Although the Luc reporter assay is important to demonstrate functionality, an appropriate negative control (mutant) derived from the 1.1 kb Ppif construct (does not bind Ppif) is needed.

We have included a mutated 1.1kb *Ppif* construct in the luc-reporter data (Figure 3D, H). The *Ppif* reporter mutagenesis luc-signal is also directly compared to other means used to prevent Smad1 binding (Supplementary Figure 1C).

5) Figure 4: The data describing Mt integrity are not very informative and should be supplemented with mitochondrial function assays.

There are several papers describing mitochondrial function in both undifferentiated osteprogenitors and differentiated osteoblasts, including publications from our group^(1, 6-8)^. Pore activity assay is used in Figure 4 to demonstrate and confirm that CypD downregulation is accompanied by a decreased MPTP activity in a dose response manner during osteogenic differentiation. To include more informative data, we compared mitochondrial function in either BMSCs and osteoblasts to their respective pairs overexpressing CypD (Figure 6G-K).

Reviewer #1 (Recommendations for the authors):General Comments:1. The study focuses on BMP signaling and dismisses other osteogenic pathways such as WNT and β-catenin signaling. It would be very informative and would significantly enhance the study if the authors determine if the CypD mechanism is BMP-specific or not.

Wnt activation effect on *Ppif* activity is now included in Supplementary Figure S4.

2. Osteogenesis and mitochondrial activity vastly change during bone diseases linked to obesity, aging, etc. It would be very exciting to determine the role of CypD in a disease/aging setting to highlight the relevance of the proposed mechanism and its potential utility as a therapeutic tool. This would constitute a major advance.

CypD expression and MPTP activity are described to be upregulated in aging^(9-11)^. Here, we showed that rescuing CypD expression in vivo also rescues the bone phenotype observed in aging. Definitely, a future direction is to further study how CypD regulation affects bone aging in more detail and how to take advantage of CypD regulatory mechanism as a therapeutic intervention.

Specific comments:3. Introduction: there are several redundant statements that hinder flow and should be streamlined.

We believe this manuscript could be interesting to a very diverse audience therefore, an expanded background information might be needed.

4. The hypothesis is scantily articulated. All cells undergo OxPhos bioenergetic shift to facilitate differentiation. In fact, a generic argument can be made for other pathways that regulate OB differentiation. Nevertheless, the study does very little to address the hypothesis that "MTPTP inhibition via CypD transcriptional downregulation is required to sustain OxPhos function". The concept requires revision and better articulation

Actually, some cells undergoing differentiation do not necessarily shift to a higher OxPhos usage. Although OxPhos is needed for cell proliferation to sustain nucleotide synthesis and biomass accrual^(12)^, the maturation and terminal differentiation of some cells can lead to no changes or even a decrease in total OxPhos usage when compared to their progenitor cells, as example of endothelial cells^(13)^ and some cells from hematopoietic lineage^(14)^. In the present study, we demonstrate that Smad1 represses *Ppif* activity, on the other hand CypD overexpression impairs osteoblast function, blunts ETC flown (Figure 6G-K) and therefore OxPhos function. We also have previously shown that BMP signaling pathway activates OxPhos in osteogenic cells^(8)^, when all these data are combined with other’s groups studies from current literature (as discussed in the discussion session), they present reasonable evidence to sustain our hypothesis.

Results:5. Figure 2: rigor and presentation of data are marginal. Specificity of the binding should be tested using SBE-mutant vectors. Data should be normalized to input in both vehicle and BMP2 treated conditions.

See item #3

6. Figure 3: Although the Luc reporter assay is important to demonstrate functionality, an appropriate negative control (mutant) derived from the 1.1 kb Ppif construct (does not bind Ppif) is needed.

See item #4

7. Figure 4: The data describing Mt integrity are not very informative and should be supplemented with mitochondrial function assays.

See item #5

8. Figure 6: the CypD GOF experiments (caCypD) corroborate previous findings in terms of osteogenesis and as such offer marginal information. At minimum, OxPhos and other Mt functional properties should be characterized in the GOF model. Furthermore, cyclosporin A is a well-known CypD inhibitor which can be used to validate this system. Despite these confirmatory findings, it remains unclear how this information advances current knowledge, which is judged as incremental (please refer to general comments).

This study presents for the first time a model for CypD GOF in osteogenic cells in vitro and in vivo. Assuming that CypD overexpression would show opposite effects to a CypD loss-of-function model is tempting, but it is not necessarily true. CypD expression is tissue and cell specific not only in terms of CypD levels but also regarding the effects of CypD on cell function^(15)^. Contrary to what we showed in this study using osteogenic cells and other studies in cardiomyocytes and neuron cells, some other cell types require higher levels of CypD expression for proper functioning. Additionally, some cells and conditions present very low sensitivity to CypD increased expression, meaning that even higher levels of CypD do not exert any effect after a determined threshold^(16)^, or can even be protective against apoptotic pathways conferring augmented cell survival^(17)^. Therefore this study is instrumental in demonstrating that CypD overexpression is in fact detrimental for osteogenic lineage, on top of characterizing the role of R-Smads in CypD transcriptional regulation. Even though CsA is vastly studied as an inhibitor of CypD, it lacks specificity affecting other signaling pathways important for osteogenic lineage, its mechanism of action is not totally understood^(18, 19)^, and it would be impossible to determine the cell-autonomous effect of CypD regulation on osteoblasts in vivo*.* Not surprisingly, CsA use in clinical trials as CypD inhibitor is mostly inconclusive, and presents several failed Phase III clinical trials, as addressed in the Discussion section.

In sum, the study requires major revision including additional experiments avoiding confirmatory data. The general role of CypD in osteogenesis is already known, yet its utility in pathologic bone loss and additional insights in its transcriptional regulation if developed further, to address novel aspects, would fill important knowledge gaps and elevate significance of this research.Reviewer #2 (Recommendations for the authors):General comments: The authors propose a novel hypothesis about the regulation of cell function by BMP signaling and here use bone cells as a model system to test this hypothesis. While data obtained from individual experiments are convincing, overall, several conclusions require additional data or clarification and there are some missed opportunities to strengthen the manuscript.Specific comments:1. Previous published work by the authors demonstrated that loss of cyclophilin D on Prx1+ cells did not have an effect on bone cell function and on bone mass in the limb skeleton while global deletion of cyclophilin D had strong negative consequences on bone mass and fracture healing. These findings are not easily compatible with the authors contention that BMP signaling in bone forming cells regulates mitochondrial activity, as most if not all bone forming cells arise from Prx1+ progenitors. It would be helpful if the authors could provide a stronger explanation for these disparities.

See item #2

2. BMP activity in bone is known to decline with age, consistent with the idea that reduced BMP signaling may be a cause of age-related bone loss. The authors have an opportunity to directly examine BMP signaling in the aging skeleton and determine if reduced signaling correlated with mitochondrial pore opening and loss of bone cell function.

It is a very interesting suggestion for a future study

3. The disease FOP is a gain of function of BMP signaling. The authors might consider examining the mouse model of FOP to see if there is a change in the mitochondrial physiology/pore opening in this disease.

It is a very interesting suggestion as a future direction as well.

4. Other pathways are important regulators of bone formation. It would be interesting to know if cyclophilin D is regulated by Wnt signaling, or if the regulation is BMP exclusive.

See item #1

Reviewer #3 (Recommendations for the authors):This is a relatively well done study which uses promoter analysis for transcription factor (Smad) binding using mutational analysis of promoter-reporter constructs and ChIP analysis. They use mouse ST2 and MC3T3-e1 cells for in vitro studies and confirm most of these in vitro findings using CypD null mice. In general, this manuscript should be acceptable for publication in eLife.The main concern is that the studies may not address the wide enormity of this pathway with multiple BMPs and Smads affecting an array of physiological process. For example, do the Ppif knock down or use of Smad inhibitor affect only the osteoblastic pathway or other skeletal-muscular pathways? The second point is that the transcription pathway is active in a basal level suggesting that there may be some dose-effect in the modulation of different physiological pathway. The authors do not address this issue.

See item #1. We included a dose-response assay to assess the effect of different concentrations of BMP2 on *Ppif* activity. We agree that there is an enormity of multiple BMP ligands and Smad interactions that can play a role in transcriptional regulation. But, we believe we took a solid step in unrevealing that BMP/Smad1-dependent is a repressor of *Ppif* activity in osteogenic cells. We are also confident that this study can encourage others to further explore other pathways, interactions, and cell types in the transcriptional regulation of CypD.

Most of the reagents are commercially available and this is not an issue.

References

1. Shum LC, White NS, Nadtochiy SM, Bentley KL, Brookes PS, Jonason JH, et al. Cyclophilin D Knock-Out Mice Show Enhanced Resistance to Osteoporosis and to Metabolic Changes Observed in Aging Bone. PLoS One. 2016;11(5):e0155709.

2. Shares BH, Smith CO, Sheu TJ, Sautchuk R, Jr., Schilling K, Shum LC, et al. Inhibition of the mitochondrial permeability transition improves bone fracture repair. Bone. 2020;137:115391.

3. Baker CE, Moore-Lotridge SN, Hysong AA, Posey SL, Robinette JP, Blum DM, et al. Bone Fracture Acute Phase Response-A Unifying Theory of Fracture Repair: Clinical and Scientific Implications. Clin Rev Bone Miner Metab. 2018;16(4):142-58.

4. Das PM, Ramachandran K, vanWert J, Singal R. Chromatin immunoprecipitation assay. Biotechniques. 2004;37(6):961-9.

5. Yamakawa T, Itakura K. Chromatin Immunoprecipitation Assay Using Micrococcal Nucleases in Mammalian Cells. J Vis Exp. 2019(147).

6. Shum LC, White NS, Mills BN, Bentley KL, Eliseev RA. Energy Metabolism in Mesenchymal Stem Cells During Osteogenic Differentiation. Stem Cells Dev. 2016;25(2):114-22.

7. Shares BH, Busch M, White N, Shum L, Eliseev RA. Active mitochondria support osteogenic differentiation by stimulating β-catenin acetylation. J Biol Chem. 2018;293(41):16019-27.

8. Smith CO, Eliseev RA. Energy Metabolism During Osteogenic Differentiation: The Role of Akt. Stem Cells Dev. 2021;30(3):149-62.

9. Barja G. Mitochondrial oxygen radical generation and leak: sites of production in states 4 and 3, organ specificity, and relation to aging and longevity. J Bioenerg Biomembr. 1999;31(4):347-66.

10. Sun N, Youle RJ, Finkel T. The Mitochondrial Basis of Aging. Mol Cell. 2016;61(5):654-66.

11. Rottenberg H, Hoek JB. The path from mitochondrial ROS to aging runs through the mitochondrial permeability transition pore. Aging Cell. 2017;16(5):943-55.

12. Zhu J, Thompson CB. Metabolic regulation of cell growth and proliferation. Nat Rev Mol Cell Biol. 2019;20(7):436-50.

13. Peng H, Wang X, Du J, Cui Q, Huang Y, Jin H. Metabolic Reprogramming of Vascular Endothelial Cells: Basic Research and Clinical Applications. Front Cell Dev Biol. 2021;9:626047.

14. Loftus RM, Finlay DK. Immunometabolism: Cellular Metabolism Turns Immune Regulator. J Biol Chem. 2016;291(1):1-10.

15. Laker RC, Taddeo EP, Akhtar YN, Zhang M, Hoehn KL, Yan Z. The Mitochondrial Permeability Transition Pore Regulator Cyclophilin D Exhibits Tissue-Specific Control of Metabolic Homeostasis. PLoS One. 2016;11(12):e0167910.

16. Bernardi P, Rasola A, Forte M, Lippe G. The Mitochondrial Permeability Transition Pore: Channel Formation by F-ATP Synthase, Integration in Signal Transduction, and Role in Pathophysiology. Physiol Rev. 2015;95(4):1111-55.

17. Eliseev RA, Malecki J, Lester T, Zhang Y, Humphrey J, Gunter TE. Cyclophilin D interacts with Bcl2 and exerts an anti-apoptotic effect. J Biol Chem. 2009;284(15):9692-9.

18. Zulian A, Rizzo E, Schiavone M, Palma E, Tagliavini F, Blaauw B, et al. NIM811, a cyclophilin inhibitor without immunosuppressive activity, is beneficial in collagen VI congenital muscular dystrophy models. Hum Mol Genet. 2014;23(20):5353-63.

19. Sileikyte J, Forte M. Shutting down the pore: The search for small molecule inhibitors of the mitochondrial permeability transition. Biochim Biophys Acta. 2016;1857(8):1197-202.

[Editors’ note: further revisions were suggested prior to acceptance, as described below.]

Reviewer #2 (Recommendations for the authors):The authors have not satisfactorily addressed my major concerns, reiterated below:Prior publications by the authors reported that global deletion of cypD resulted in no obvious skeletal phenotype until mice began to age when wt mice lost bone mass and cypD null mice did not undergo age related bone loss. Next they reported that global loss of cypD enhanced fracture repair in older mice. They later published that loss of cypD in Prx1+ cells did not positively affect bone function stating "lineage deletion of cypD is not sufficient to improve fracture repair," and that it was the systemic loss of cypD that affected the skelton. These data raise 3 issues:a) In mice, Prx1+ cells make up the majority of cells forming the fracture callus and new bone, making it difficult to understand what osteolineage-specific effects are regulated by cypD.

Indeed, it is important to connect our new data with previous findings and we do not think there is a discrepancy. In the current manuscript we are not focusing on fracture repair and not using Prx1-Cre driver. Fracture repair involving endochondral ossification and multiple interactions is mechanistically different from bone maintenance and aging which is our focus here. We believe that our current data on bone aging in Col1-Cre-driven CypD deletion in fact complement our prior work with global CypD deletion. We have observed similar results here. Moreover, CypD gain-of-function in osteoblasts using AAV-Cre-DIO vector, reversed the anti-aging effect of CypD deletion confirming the role of CypD in osteolineage aging.

With regards to Prx1, it indeed labeled a lot of cells in the fracture callus as we showed in our ’20 Bone publication. A possible reason why CypD deletion from Prx1+ cells did not stimulate repair is as follows: we are presenting new data on fracture showing that BMP2 levels are upregulated while CypD expression is downregulated during fracture repair (Figure 7 —figure supplement 1D, E). Publicly available gene expression data sets on fracture repair show similar findings (Figure 7 —figure supplement 1A, top right table). These findings may explain why we found no effects of CypD conditional deletion in Prx1-expressing osteoprogenitors. Since in young mice with functional BMP signaling, CypD is downregulated during fracture repair, the effect of any additional bone-specific CypD downregulation may be blunted.

b) Fracture repair cannot occur in the absence of BMP signaling in Prx1+ cells; if BMP signaling is an important physiological regulator of cypD in osteolineage cells, there should be a relationship between the loss of BMP signaling and loss of cypD in this population.

Here again we would like to refrain from adding a detailed discussion of our previous data using fracture and Prx1-Cre for the reasons stated above in our response (a). However, we appreciate the point about the relationship between changes in BMP signaling and CypD expression. This can definitely make our work stronger and more significant. We, therefore, analyzed available data on fracture and on bone aging and found inverse correlation between BMP signaling and CypD expression.

For example, Ambrosi et al., (Nature. 2021, 597: 256–262. Data deposited at GSE166441) performed RNAseq of young and aged bones and found (Figure 7 —figure supplement 1A, top left table) that BMP-dependent *Smad1* was decreased while CypD gene, *Ppif*, increased in aged bone. On the other hand, Coates et al., studied changes in fracture callus gene expression (Bone 127 (2019) 577–591, Data deposited at GSE152677). Analysis of their data showed that on Day 14 post fracture and when compared to the unfractured bone, BMP-dependent *Smad 1* increased while *Ppif* decreased (Figure 7 —figure supplement 1A, top right table). In conditions leading to pathological calcification such as osteoarthritis and osteophyte formation (Dunn, Soul et al., 2016) and cardiac valve calcification (Huang, Xu et al., 2019) and associated with upregulated BMP/Smad signaling, *PPIF* expression is decreased (Figure 7 —figure supplement 1A, bottom tables).

We also analyzed bone samples from 3 and 18 month old mice that were collected in our previous work. Immunofluorescent staining of bones from these mice showed decreased levels of BMP2 with correspondingly increased levels of CypD in bones of aged mice when compared to young mice (Figure 7 —figure supplement 1B, C). We also used bone fracture samples from our previous work and performed immunofluorescence for BMP2 and CypD. Consistent with the published RNAseq data set described above, BMP2 signal significantly increased while CypD signal significantly decreased in the callus at Day 14 post fracture when compared to the unfractured bone (Figure 7 —figure supplement 1D, E). This increase likely reflects active bone forming and callus ossification stage. These new data were added to the Results section (page 11) and discussed in the Discussion section (page 19).

Overall, previously published transcriptomic data and our own studies indicate inverse correlation between BMP signaling and CypD expression.

c) Without addressing the in vivo models suggested (FOP, aging,) and demonstrating that changes in BMP signaling in vivo in a physiologically relevant model result in changes in cypD in osteolineage cells, the study does not reach a level of physiological significance.

This is an excellent comment and we are, in fact, planning to further elucidate the connection between BMP signaling, CypD, and CypD-dependent mitochondrial integrity in our future work. This will involve mouse models of gain- and loss-of-function of BMP signaling in osteolineage cells. We believe that the significance of the current work is several fold: (1) we have demonstrated for the first time that BMP signaling acts as a transcriptional regulator of CypD; (2) we have shown the ‘pro-aging’ role of CypD in osteolineage cells in vitro and in vivo; and (3) our data and previously published data sets strongly indicate inverse relationship between BMP signaling and CypD expression.

Since the authors have utilized a variety of mouse models and have not reported the complete phenotype of any, I suggest including a table that clearly delineates and compares the specific skeletal manifestations of cypD genetic deletion, cypD pharmacological deletion, and lineage-specific cypD deletion. Bone mass, bone strength and ability to repair fractures should be included as well as levels of BMP signaling in the skeletal tissues that demonstrate a phenotype.

We indeed do not have all the suggested skeletal data and what we have was published as separate works. For instance, our 2016 PLoS One paper (doi.org/10.1371/journal.pone.0155709) has data on the effect of global CypD KO on bone maintenance in aging with micro-CT and biomechanical assays. Our 2020 Bone paper (doi.org/10.1016/j.bone.2020.115391) has data on the effect of global CypD KO, systemic CypD inhibitor, or Prx1-Cre driven CypD cKO on fracture repair (micro-CT, biomechanical, etc). We are including a summary of our findings in a Table (Figure 7—figure supplement 3), while detailed data are presented in the listed publications.